# TiG-BEV: Multi-view BEV 3D Object Detection via Target Inner-Geometry Learning

## Abstract

To achieve accurate multi-view 3D object detection, existing methods propose to benefit camera-based detectors with spatial cues provided by the LiDAR modality, e.g., depth supervision and bird-eye-view (BEV) feature distillation. However, they employ a direct point-to-point mimicry from LiDAR to camera, which suffers from the modality gap between Camera-LiDAR features. In this paper, we propose the **T**arget **I**nner-**G**eometry learning scheme to enhance camera-based BEV detectors from both depth and BEV feature by leveraging the LiDAR modality, termed as **TiG-BEV**. Firstly, we introduce an inner-depth supervision module to learn the low-level relative depth relations in each object. This equips camera-based detectors with a deeper understanding of object-level spatial structures. Secondly, we design an inner-feature BEV distillation module to imitate the high-level semantics of different keypoints within foreground targets. To further alleviate the domain gap between two modalities, we incorporate both inter-channel and inter-keypoint distillation to model feature similarity. With our target inner-geometry learning, TiG-BEV effectively boosts BEVDepth by +2.3% NDS on nuScenes val set, and achieves leading performance with 61.9% NDS on nuScenes leaderboard.

## 1 Introduction

3D object detection aims to recognize and localize objects in 3D space, which has made remarkable strides in various applications, such as robotics (Antonello et al., 2017) and autonomous driving (Shi et al., 2020; Wang et al., 2021b; Chen et al., 2022a; Caesar et al., 2020). Mainstream methods for 3D object detection can be categorized into LiDAR-based detectors (Deng et al., 2021; Shi et al., 2020; 2019) and camera-based detectors (Li et al., 2022b;c; Wang et al., 2021a; 2022b; Zhang et al., 2022a;b). LiDAR-based methods achieve outstanding performance by taking 3D point clouds as input, which inherently contains rich spatial structures. In contrast, camera-based methods, while providing cost-effective color context, are limited by the absence of geometric depth cues.

To address these performance disparities, existing methods leverage the spatial cues provided by the LiDAR modality to enhance the precision of camera-based detectors. This enhancement primarily falls into two schemes, as illustrated in Figure 1. (1) Dense Depth Supervision (Figure 1 (a)), e.g., CaDDN (Reading et al., 2021) and BEVDepth (Li et al., 2022b). These methods project the input LiDAR points onto image planes as depth maps, and explicitly supervise the categorical depth prediction within both foreground and background regions. (2) BEV Feature Distillation (Figure 1 (b)), e.g., CMKD (Hong et al., 2022) and BEVDistill (Chen et al., 2022b). Employing the teacher-student paradigm, these methods compel the camera-based detector (the student) to imitate the BEV representation of a pre-trained LiDAR-based detector (the teacher). By directly mimicking the BEV features, the student inherits the encoded high-level BEV semantics from the teacher.

However, existing methods fall short in capturing the inner-geometric characteristics of foreground targets. Inner geometry of an object includes its low-level spatial contours and high-level part-wise semantic relations, which are significant for precise object recognition and localization. For instance, BEVDepth simply employs pixel-level depth supervision without tailoring it to capture relative depth within objects, and BEVDistill applies foreground-guided distillation but neglects the inner relations of BEV features. In addition, methods (Chen et al., 2022b; Hong et al., 2022) for BEV feature distillation directly enforce the channel-level alignment between cross-modal BEV represen-

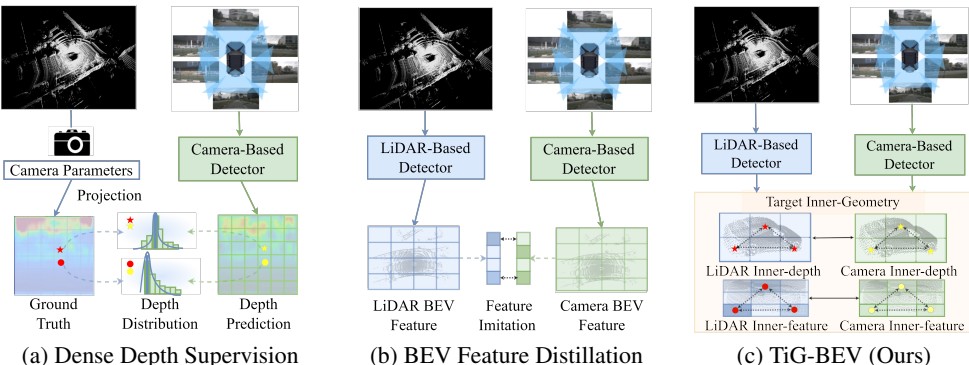

Figure 1: **Different LiDAR-to-Camera Learning Schemes: (a) Dense Depth Supervision (Li et al., 2022b; Reading et al., 2021)**, which directly supervises the categorial depth distribution of every valid pixel in the whole depth map, **(b) BEV Feature Distillation (Chen et al., 2022b; Hong et al., 2022)**, which constrainedly aligns the value of BEV feature between different modalities, **(c) Our Target Inner-Geometry Learning (TiG-BEV)**, which utilizes both the low-level inner-depth relations and the high-level inner-feature semantics of foreground targets.

tations. Such strict mimicry might have a detrimental impact on performance due to the modality gap between camera and LiDAR BEV features, i.e., visual appearances vs. spatial geometries.

To alleviate this issue, we propose a novel LiDAR-to-camera learning scheme, **TiG-BEV**, which involves the inner-geometry of foreground targets into the camera-based detectors for multi-view BEV 3D object detection. As shown in Figure 1 (c), we simultaneously perform target inner-geometry learning for both depth prediction and BEV representation learning. Firstly, besides the previous absolute depth map prediction (Li et al., 2022; Reading et al., 2021), we introduce an inner-depth supervision module within pixels of different foreground targets. A reference point of depth is adaptively selected for each target to obtain the relative depth relationships shown in Figure 2, which contributes to high-quality depth map prediction with better target structural understanding. Secondly, we propose an inner-feature BEV distillation module, which imitates the high-level foreground BEV semantics generated by a pre-trained LiDAR-based detector. Different from previous methods of dense and strict feature distillation (Chen et al., 2022b; Hong et al., 2022), we adaptively sample several keypoints within each BEV foreground area and guide the camera-based detector to learn their inner feature-similarities shown in Figure 3, which are in both inter-channel and inter-keypoint manners. This approach enables the camera-based detector to not only inherit high-level, part-wise LiDAR semantics but also relieve the modality gap by avoiding strict feature mimicry. Our extensive experiments consistently demonstrate performance improvements achieved by TiG-BEV compared to the baseline models.

The contributions of TiG-BEV are summarized below:

- We introduce an inner-depth supervision module that enables us to capture the internal depth relations between different parts of each foreground target and leads to a better target depth map prediction which is important to the perspective transformation to obtain the BEV feature.
- We propose an inner-feature BEV distillation module to transfer the well-learned knowledge from LiDAR modality to camera-based BEV representations with the inner-geometry learning for high-level BEV semantics instead of directly feature alignment.
- Extensive experiments have confirmed our effectiveness to enhance multi-view BEV 3D object detection. On nuScenes val set, the powerful BEVDepth is boosted by **+2.3%** NDS and **+2.4%** mAP, and with further enhancements of **+3.0%** NDS and **+4.1%** mAP on test set.

## 2 RELATED WORK

**Camera-based 3D Object Detection.** Camera-based 3D object detection has been widely used for applications like autonomous driving since its low cost compared with LiDAR-based detec-

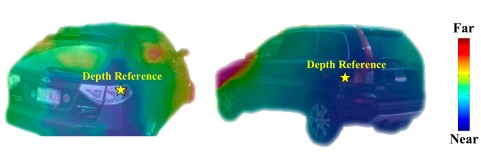

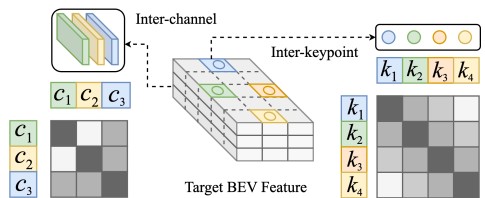

Figure 2: **Inner-depth Supervision.** We guide the camera-based detector to learn the relative spatial structures within the target foreground areas. A depth reference point (dotted in yellow) is adaptively selected to calculate relative depth values.

Figure 3: **Inner-feature BEV Distillation.** We respectively conduct inter-channel and inter-keypoint feature distillation in BEV space for the camera-based detector, which alleviates the cross-modal semantic gap and boosts inner-geometry learning.

tors. FCOS3D (Wang et al., 2021a) first predicts the 3D attributes of objects through the features around 2D centers and MonoDETR (Zhang et al., 2022a) introduces DETR-like (Carion et al., 2020) architectures without complex post-processing. Recently, Bird's-Eye-View (BEV), as a unified representation of surrounding views, has attracted much attention. DETR3D (Wang et al., 2022b) and PETR series(Liu et al., 2022a;b) adopt 3D object queries and integrate 3D reference points for feature aggregation. BEVDet series (Huang et al., 2021; Huang & Huang, 2022) utilize LSS (Philion & Fidler, 2020) to transform 2D image features into 3D BEV representation. BEVFormer (Li et al., 2022c) automates the camera-to-BEV process with learnable attention modules and BEV queries in 3D space. BEVDepth (Li et al., 2022b) observes that accurate depth estimation is essential for BEV 3D object detection supervised by projected LiDAR points. We defer more related works in Appendix A due to the space constraints. As a LiDAR-to-camera learning scheme, our TiG-BEV leverages the pre-trained LiDAR-based detector to improve the performance of camera-based detectors for multi-view BEV 3D object detection.

## 3 METHOD

The overall architecture of TiG-BEV is shown in Figure 4, which consists of three components: the student camera-based detector, the teacher LiDAR-based detector, and our proposed target inner-geometry learning scheme. In Section 3.1, we first introduce the adopted baseline models. Then, we specifically illustrate how TiG-BEV distills the inner-geometry characteristics, including inner-depth supervision (Section 3.2) and inner-BEV feature distillation (Section 3.3). Finally in Section 3.4, we present the overall loss of for LiDAR-to-camera learning.

### 3.1 BASELINE MODELS

**Student Camera-based Detector.** By default, we adopt BEVDepth (Li et al., 2022b) as our student camera-based detector for multi-view 3D object detection. Given the input multi-view images (normally 6 views for a scene), the student model first utilizes a shared 2D backbone and FPN module (Lin et al., 2017) to extract the $C$-channel visual features $\{F_i\}_{i=1}^6$, where $F_i \in \mathbb{R}^{C \times H_v \times W_v}$, and $H_v, W_v$ denote the size of feature maps. These features are fed into a shared depth network to generate the categorical depth map (Reading et al., 2021), $\{D_i\}_{i=1}^6$, where $D_i \in \mathbb{R}^{D \times H_v \times W_v}$, where D denotes the pre-defined number of depth bins. During training, BEVDepth adopts dense depth supervision for the predicted depth maps, which projects the paired LiDAR input onto multi-view image planes to construct pixel-by-pixel absolute depth ground truth, $\{D_i^{gt}\}_{i=1}^6$, where $D_i^{gt} \in \mathbb{R}^{1 \times H_v \times W_v}$. Then, following (Philion & Fidler, 2020), the multi-view visual features are projected into a unified BEV representation via the predicted depth maps, which is further encoded by a BEV encoder, denoted as $F_{\text{bev}}^{cam} \in \mathbb{R}^{C \times H_{\text{bev}} \times W_{\text{bev}}}$. Finally, the detection heads are applied on top to predict objects in 3D space. We represent the two basic losses of the student model as $\mathcal{L}_{\text{depth}}^A$ and $\mathcal{L}_{\text{det}}$, respectively denoting the Binary Cross Entropy loss for dense absolute depth values and the 3D detection loss.

**Teacher LiDAR-based Detector.** We select the popular LiDAR detector CenterPoint (Yin et al., 2021) as the teacher for target inner-geometry learning. Given the input point cloud data, CenterPoint voxelizes into grid-based data and utilizes a 3D backbone to obtain the $C$-channel LiDAR

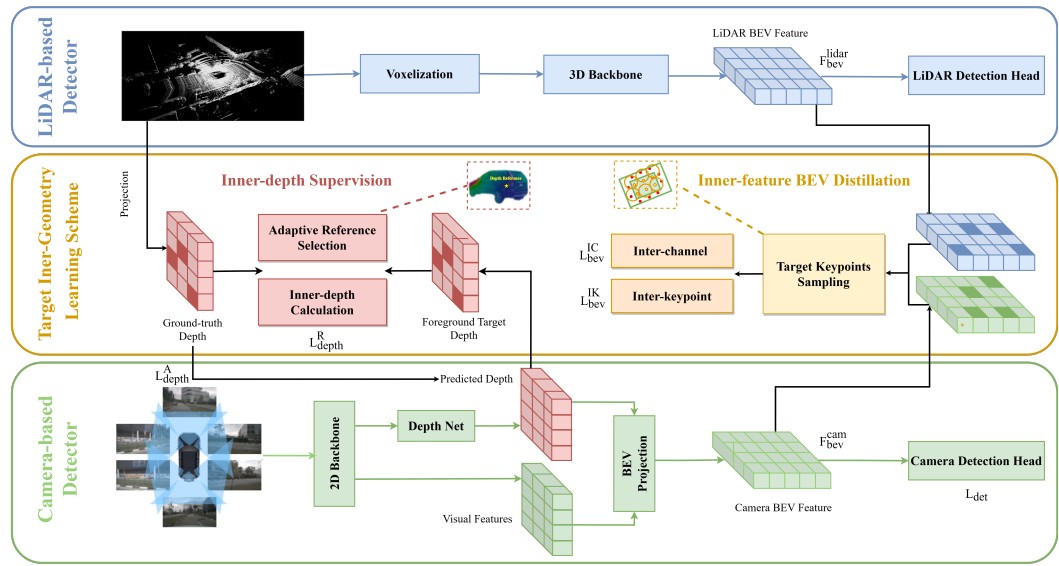

Figure 4: **Overall Framework of TiG-BEV,** which contains a pre-trained LiDAR-based detector as teacher, a camera-based detector as student, and a target inner-geometry scheme for cross-modal learning. Our proposed learning paradigm transfers the inner-geometry semantics of the LiDAR modality via two components, an inner-depth supervision (Section 3.2) for foreground relative depth, and an inner-feature BEV distillation (Section 3.3) from both channel-wise and keypoint-wise.

BEV feature $F_{\text{bev}}^{lidar} \in \mathbb{R}^{C \times H_{\text{bev}} \times W_{\text{bev}}}$, which has the same feature size as $F_{\text{bev}}^{cam}$ from the student detector. As the CenterPoint has been well pre-trained, $F_{bev}^{lidar}$ can provide the student BEV feature with sufficient geometric and semantic knowledge, espeically in the target foreground areas. Note that the LiDAR-based teacher is merely required during training for cross-modal learning, and for inference, only multi-view images are token as input for the camera-based detector.

## 3.2 INNER-DEPTH SUPERVISION

In addition to the dense absolute depth supervision, we propose to guide the student model to learn the inner-depth geometries in different target foreground areas. As shown in Figure 5 (a), for the instance level, the existing absolute depth supervision with categorical representation ignores the relative structural information inside each object and provide no explicit fine-grained depth signals. Therefore, we propose to additionally conduct inner-depth supervision with continuous values from the LiDAR projected depth maps shown in Figure 5 (b), which effectively boosts the network to capture the inner-geometry of object targets.

**Foreground Target Localization.** To accurately obtain the inner-depth values, we first localize the foreground pixels for each object targets in the depth maps. Given the ground-truth 3D bounding boxes, we extract the corresponding 3D LiDAR points inside the box for each object target, and project them onto different image planes. In this way, we can attain the pixels within foreground object areas on both the predicted and ground-truth depth maps, $\{D_i, D_i^{gt}\}_{i=1}^6$. The foreground pixels can roughly depict the geometric contour of different target objects and well improve the subsequent inner-depth learning. We taking the $i$-th view as an example and omit the index $i$ in the following texts for simplicity. Suppose there exist $M$ target objects on the image, we denote the foreground depth-value set for the $M$ objects as $\{S_j, S_j^{gt}\}_{j=1}^M$, where each $\{S_j, S_j^{gt}\}$ includes the foreground categorical depth prediction and ground-truth depth values for the $j$-th target.

**Continuous Depth Representation.** Different from the categorical representation of absolute depth values, we represent the predicted inner depth of foreground targets by continuous values, which reflects more fine-grained geometric variations. For pixel $(x, y)$ of the $j$-th target object $S_j$, the predicted possibility of $k$-th depth bin is denoted as $S_j(x, y)[k]$, where $1 \leq k \leq D$. Then, we

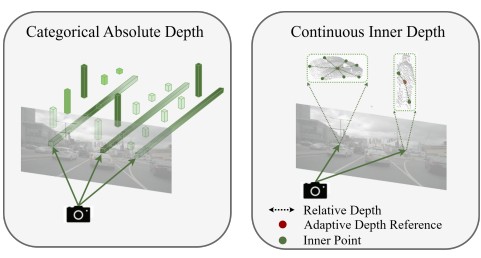 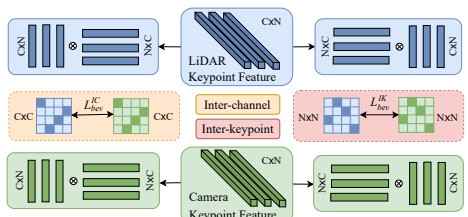

Figure 5: **Comparison of Categorical Absolute Depth and Continuous Inner Depth.** We adopt the inner-depth supervision with continuous depth values to guide the camera-based student to learn local spatial structures of foreground object targets.

Figure 6: **Details of Innter-feature BEV Distillation.** For each foreground area in BEV space, we represent rach target feature by a set of keypoints and conduct feature distillation in both inter-channel and inter-keypoint manners.

calculate the continuous depth value $\hat{S}_j(x, y)$ for the pixel $(x, y)$ as

$$\hat{S}_j(x, y) = \sum_{k=1}^{D} (\hat{S}[k] \cdot S_j(x, y)[k]), \tag{1}$$

where $\hat{S}[k]$ denotes the depth value of the $k$-th bin center. By this, we convert the categorical depth prediction of different target objects, $\{S_j\}_{j=1}^{M}$, into continuous representations, denoted as $\{\hat{S}_j\}_{j=1}^{M}$.

**Adaptive Depth Reference.** To calculate the relative depth values, we propose to utilize an adaptive depth reference for different foreground targets. Specifically, according to the predicted continuous depth values in $\{\hat{S}_j\}_{j=1}^{M}$, we select the pixel with the smallest depth prediction error as the reference point for each target, and correspondingly set its depth value as the depth reference, as shown in Figure 5. For the $j$-th target with the ground-truth inner-depth $\{\hat{S}_j, \hat{S}_j^{gt}\}_{j=1}^{M}$, we calculate the depth reference point $(x_r, y_r)$ by

$$(x_r, y_r) = \underset{(x,y) \in \hat{S}_j}{\text{Argmin}} \left( \hat{S}_j^{gt}(x, y) - \hat{S}_j(x, y) \right). \tag{2}$$

Then, the predicted and ground-truth reference depth values are denoted as $\hat{S}_j(x_r, y_r)$ and $\hat{S}_j^{gt}(x_r, y_r)$, respectively. By adaptively selecting the reference point with the smallest error, the inner-depth distribution can dynamically adapt to objects with different shapes and appearances, which stabilizes the learning for some truncated and occluded objects.

**Inner-depth Calculation.** On top of the reference depth value, we calculate the relative depth values within the foreground area of each target object. For pixel $(x, y)$ of the $j$-th target $\{\hat{S}_j, \hat{S}_j^{gt}\}$, the predicted and ground-truth inner-depth values are formulated as

$$\begin{aligned} r\hat{S}_j(x, y) &= \hat{S}_j(x, y) - \hat{S}_j(x_r, y_r), \\ r\hat{S}_j^{gt}(x, y) &= \hat{S}_j^{gt}(x, y) - \hat{S}_j^{gt}(x_r, y_r). \end{aligned} \tag{3}$$

We denote the obtained predicted and ground-truth inner-depth value sets for $M$ target objects as $\{\hat{R}_j, \hat{R}_j^{gt}\}_{j=1}^{M}$. Finally, we supervise the inner-depth prediction by an L2 loss, formulated as

$$\mathcal{L}_{\text{depth}}^{R} = \sum_{j=1}^{M} ||\hat{R}_j - \hat{R}_j^{gt}||_2. \tag{4}$$

### 3.3 INNER-FEATURE BEV DISTILLATION

Besides the depth supervision for low-level spatial information, our TiG-BEV also adopts the inner-geometry learning for high-level BEV semantics from pre-trained LiDAR-based detectors. Previous works (Chen et al., 2022b; Hong et al., 2022) for BEV distillation directly force the student to imitate

the teacher's features point-to-point in the BEV space. In spite of the performance improvement, such strategies are constrained by the following two aspects. On the one hand, due to the sparsity of scanned point clouds, the LiDAR-based BEV features might contain redundant and noisy information in the background areas. Although BEVDistill (Chen et al., 2022b) utilizes foreground masks to alleviate this issue, such dense feature distillation still cannot provide focused and effective guidance to the student network. On the other hand, the camera-based and LiDAR-based BEV features depict different characteristics of the scene, respectively, visual appearances and spatial structures. Therefore, forcing the BEV features to be completely consistent between two modalities is sub-optimal considering the semantic gap. In our TiG-BEV, we propose an inner-feature BEV distillation (Figure 6) consisting of inter-channel and inter-keypoint learning schemes, which conducts attentive target features distillation and relieve the cross-modal semantic gap.

**Target Keypoint Extraction.** To distill the knowledge of LiDAR-based detectors only within sparse foreground regions, we extract the BEV area of each object target and represent it by a series of keypoint features. Given the ground-truth 3D bounding box for each target, we first enlarge the box size for a little bit in the BEV space to cover the entire foreground area, e.g., object contours and edges. Then, we uniformly sample its BEV bounding box by $N$ keypoints, and adopt bilinear interpolation to obtain the keypoint features from the encoded BEV representations. From both camera-based $F_{\text{bev}}^{cam}$ and LiDAR-based $F_{\text{bev}}^{lidar}$, we respectively extract the keypoint features for all $M$ object targets as $\{f_j^{cam}, f_j^{lidar}\}_{j=1}^{M}$, where $f_j^{cam}, f_j^{lidar} \in \mathbb{R}^{N \times C}$. By these, such BEV keypoints can well represent the part-wise features and the inner-geometry semantics of foreground targets.

**Inter-channel BEV Distillation.** Taking the $j$-th object target as an example, we first apply an inter-channel BEV distillation, which guides the student keypoint features to mimic the channel-wise relationships of the teacher's. Such inter-channel signals imply the overall geometric semantics of each object target. Compared with the previous channel-by-channel supervision, our inter-channel distillation can preserve the distinctive aspects of the two modalities, while effectively transfer the well pre-trained knowledge of LiDAR-based detectors. Specifically, we calculate the inter-channel similarities of both camera-based and LiDAR-based keypoint features, formulated as

$$A_j^{cam} = f_j^{cam} f_j^{cam\top}; \quad A_j^{lidar} = f_j^{lidar} f_j^{lidar\top}, \tag{5}$$

where $A_j^{cam}, A_j^{lidar} \in \mathbb{R}^{C \times C}$ denote the feature relationships between different $C$ channels for the two modalities. For all $M$ objects in a scene, we adopt L2 loss between the two inter-channel similarities for feature distillation as

$$\mathcal{L}_{\text{bev}}^{IC} = \sum_{j=1}^{M} ||A_j^{lidar} - A_j^{cam}||_2. \tag{6}$$

**Inter-keypoint BEV Distillation.** The inter-channel distillation guides the camera-based detector to learn the channel-wise diversity from the LiDAR-based teacher. However, it is conducted without considering the inner correlation of different keypoints within each object target, which is not capable of capturing the local geometries among different foreground parts, e.g., the front and rear of cars. To this end, we propose to utilize the inter-keypoint correlations of LiDAR-based BEV features and transfer such inner-geometry semantics into camera-based detectors. Analogous to the aforementioned inter-channel module, for the $j$-th target object, we calculate the inter-keypoint similarities in a transposed manner for the two modalities as

$$B_j^{cam} = f_j^{cam\top} f_j^{cam}; \quad B_j^{lidar} = f_j^{lidar\top} f_j^{lidar}, \tag{7}$$

where $B_j^{cam}, B_j^{lidar} \in \mathbb{R}^{N \times N}$ denote the feature relationships between different $N$ keypoints respectively for camera and LiDAR. We also adopt L2 loss for all $M$ targets as

$$\mathcal{L}_{\text{bev}}^{IK} = \sum_{j=1}^{M} ||B_j^{lidar} - B_j^{cam}||_2. \tag{8}$$

Then, the distillation loss for inter-channel and inter-keypoint features is formulated as

$$\mathcal{L}_{\text{bev}} = \mathcal{L}_{\text{bev}}^{IC} + \mathcal{L}_{\text{bev}}^{IK}, \tag{9}$$

where the two terms are orthogonal respectively for the channel-wise feature diversity and keypoint-wise semantic correlations.

Table 1: **Performance Comparison on nuScenes (Caesar et al., 2020) Val Set.** 'C' and 'L' denote the camera-based and LiDAR-based methods, which refer to the input data during inference. * denotes our implementation using BEVDet (Huang et al., 2021) codebase.

| Method | Modality | Backbone | Resolution | mAP↑ | NDS↑ | mATE↓ | mASE↓ | mAOE↓ | mAVE↓ | mAAE↓ |
|---|---|---|---|---|---|---|---|---|---|---|
| CenterPoint (Yin et al., 2021) | L | VoxelNet | - | 0.564 | 0.646 | 0.299 | 0.254 | 0.330 | 0.286 | 0.191 |
| FCOS3D (Wang et al., 2021a) | C | ResNet-101 | 900 × 1600 | 0.343 | 0.415 | 0.725 | 0.263 | 0.422 | 1.292 | 0.153 |
| PGD (Wang et al., 2022a) | C | ResNet-101 | 900 × 1600 | 0.369 | 0.428 | 0.683 | 0.260 | 0.439 | 1.268 | 0.185 |
| MonoDETR (Zhang et al., 2022a) | C | ResNet-101 | 900 × 1600 | 0.372 | 0.434 | 0.676 | 0.258 | 0.429 | 1.253 | 0.176 |
| DETR3D (Wang et al., 2022b) | C | ResNet-101 | 900 × 1600 | 0.303 | 0.374 | 0.860 | 0.278 | 0.437 | 0.967 | 0.235 |
| PETR (Liu et al., 2022a) | C | ResNet-101 | 512 × 1408 | 0.357 | 0.421 | 0.710 | 0.270 | 0.490 | 0.885 | 0.224 |
| BEVFormer (Li et al., 2022c) | C | ResNet-101 | 900 × 1600 | 0.416 | 0.517 | 0.673 | 0.274 | 0.372 | 0.394 | 0.198 |
| PETRv2 (Liu et al., 2022b) | C | ResNet-101 | 640 × 1600 | 0.421 | 0.524 | 0.681 | 0.267 | 0.357 | 0.377 | 0.186 |
| BEVDet (Huang et al., 2021) | C | Swin-T | 256 × 704 | 0.312 | 0.392 | 0.691 | 0.272 | 0.523 | 0.909 | 0.247 |
| + TiG-BEV | L → C | Swin-T | 256 × 704 | **0.334** | **0.421** | 0.647 | 0.270 | 0.554 | 0.777 | 0.211 |
|  |  |  |  | +2.2% | +2.9% | -4.4% | -0.2% | +3.1% | -13.2% | -3.6% |
| BEVDet4D (Huang & Huang, 2022) | C | ResNet-101 | 512 × 1408 | 0.370 | 0.492 | 0.667 | 0.272 | 0.475 | 0.331 | 0.182 |
| + TiG-BEV | L → C | ResNet-101 | 512 × 1408 | **0.412** | **0.520** | 0.608 | 0.271 | 0.451 | 0.341 | 0.187 |
|  |  |  |  | +4.2% | +2.8% | -5.9% | -0.1% | -2.4% | +1.0% | +0.5% |
| BEVDepth* (Li et al., 2022b) | C | ResNet-101 | 512 × 1408 | 0.416 | 0.521 | 0.605 | 0.268 | 0.455 | 0.333 | 0.203 |
| + TiG-BEV | L → C | ResNet-101 | 512 × 1408 | **0.440** | **0.544** | 0.570 | 0.267 | 0.392 | 0.331 | 0.201 |
|  |  |  |  | +2.4% | +2.3% | -3.5% | -0.1% | -6.3% | -0.2% | -0.2% |

## 3.4 OVERALL LOSS

To sum up, we benefit the student camera-based detector by target inner-geometry from two complementary aspects, i.e., an inner-depth supervision for low-level signals and an inner-feature BEV distillation for high-level semantics. They produce two losses as $\mathcal{L}_{\text{depth}}^{R}$ and $\mathcal{L}_{\text{bev}}$. Together with the original two losses, i.e., dense absolute depth supervision $\mathcal{L}_{\text{depth}}^{A}$, and 3D detection $\mathcal{L}_{\text{det}}$, the overall loss of our TiG-BEV is formulated as

$$\mathcal{L}_{\text{TiG}} = \mathcal{L}_{\text{det}} + \mathcal{L}_{\text{depth}}^{A} + \mathcal{L}_{\text{depth}}^{R} + \mathcal{L}_{\text{bev}}^{IC} + \mathcal{L}_{\text{bev}}^{IK}. \tag{10}$$

## 4 EXPERIMENT

In this section, we first introduce our implementation settings. Then, we conduct a series of experiments with ablation studies to show the effectiveness of TiG-BEV.

## 4.1 EXPERIMENTAL SETTINGS

**Dataset.** We evaluate our TiG-BEV on nuScenes dataset (Caesar et al., 2020), one of the most popular large-scale outdoor public datasets for autonomous driving. It consists of 700, 150, 150 scenes for training, validation and testing, respectively. We refer to Appendix B.1 for dataset details.

**Implementation Details.** We implement our TiG-BEV using the BEVDet (Huang et al., 2021; Huang & Huang, 2022) code base on 8 NVIDIA A100 GPUs, which is built on MMDetection3D toolkit (Contributors, 2020). A pre-trained CenterPoint (Yin et al., 2021) with voxel size of $[0.1, 0.1, 0.2]$ is adopted as the LiDAR-based teacher, and the camera-based students include BEVDepth (Li et al., 2022b), BEVDet (Huang et al., 2021) and BEVDet4D (Huang & Huang, 2022). During inference, the camare-based detecots only take multi-view images as input without the LiDAR data or teachers. Referring to BEVDepth, we additionally add the dense depth supervision on top of BEVDet and BEVDet4D besides our TiG-BEV. We follow their official training settings as default, including data augmentation (random flip, scale and rotation), training schedule (2x), and others (AdamW optimizer (Loshchilov & Hutter, 2017), 2e-4 learning rate and batch size 8). For nuScenes val set in Table 1, we compare our TiG-BEV with all other methods under the CBGS strategy (Zhu et al., 2019). For nuScenes test set in Table 2, we implement TiG-BEV on BEVDepth with ConvNeXt-base (Liu et al., 2022c) backbone and input images of $640 \times 1600$. We train our model for 20 epochs with CBGS and evaluate with test time augmentation. For all other results, we do not utilize CBGS to better reveal the significance of proposed methods.

Table 2: **Performance Comparison on nuScenes (Caesar et al., 2020) Test Set.** 'C' denotes the camera-based method, which refers to the input during inference. * denotes our implementation the same as BEVDistill (Chen et al., 2022b) and † is reported by DistillBEV (Wang et al., 2023).

| Method | Modality | Backbone | Resolution | mAP↑ | NDS↑ | mATE↓ | mASE↓ | mAOE↓ | mAVE↓ | mAAE↓ |
|---|---|---|---|---|---|---|---|---|---|---|
| CenterPoint (Yin et al., 2021) | L | VoxelNet | - | 0.603 | 0.673 | 0.262 | 0.239 | 0.361 | 0.288 | 0.136 |
| FCOS3D (Wang et al., 2021a) | C | ResNet-101 | 900 ×1600 | 0.358 | 0.428 | 0.690 | 0.249 | 0.452 | 1.434 | 0.124 |
| PGD (Wang et al., 2022a) | C | ResNet-101 | 900 ×1600 | 0.386 | 0.448 | 0.626 | 0.245 | 0.451 | 1.509 | 0.127 |
| DETR3D (Wang et al., 2022b) | C | VoVNet-99 | 900 ×1600 | 0.412 | 0.479 | 0.641 | 0.255 | 0.394 | 0.845 | 0.133 |
| PETR (Liu et al., 2022a) | C | VoVNet-99 | 900 ×1600 | 0.441 | 0.504 | 0.593 | 0.249 | 0.384 | 0.808 | 0.132 |
| BEVFormer (Li et al., 2022c) | C | VoVNet-99 | 900 ×1600 | 0.481 | 0.569 | 0.582 | 0.256 | 0.375 | 0.378 | 0.126 |
| PETRv2 (Liu et al., 2022b) | C | VoVNet-99 | 640 ×1600 | 0.490 | 0.582 | 0.561 | 0.243 | 0.361 | 0.343 | 0.120 |
| BEVDet (Huang et al., 2021) | C | Swin-B | 640 ×1600 | 0.424 | 0.488 | 0.524 | 0.242 | 0.373 | 0.950 | 0.148 |
| BEVDet4D (Huang & Huang, 2022) | C | Swin-B | 640 ×1600 | 0.451 | 0.569 | 0.511 | 0.241 | 0.386 | 0.301 | 0.121 |
| Unidistill (Zhou et al., 2023) | L → C | ResNet-50 | 256 ×704 | 0.289 | 0.384 | 0.659 | 0.259 | 0.514 | 1.064 | 0.170 |
| X³KD (Klingner et al., 2023) | L → C | ResNet-101 | 640 ×1600 | 0.456 | 0.561 | 0.506 | 0.253 | 0.414 | 0.366 | 0.131 |
| BEVDepth† (Li et al., 2022b) | C | Swin-B | 640 ×1600 | 0.489 | 0.590 | - | - | - | - | - |
| + DistillBEV (Wang et al., 2023) | L → C | Swin-B | 640 ×1600 | 0.525 | 0.612 | - | - | - | - | - |
| BEVDepth* (Li et al., 2022b) | C | ConvNeXt-B | 640 ×1600 | 0.491 | 0.589 | 0.484 | 0.245 | 0.377 | 0.320 | 0.132 |
| + BEVDistill (Chen et al., 2022b) | L → C | ConvNeXt-B | 640 ×1600 | 0.498 | 0.594 | 0.472 | 0.247 | 0.378 | 0.326 | 0.125 |
| **+ TiG-BEV** | L → C | ConvNeXt-B | 640 ×1600 | **0.532** | **0.619** | 0.450 | 0.244 | 0.343 | 0.306 | 0.132 |
| | | | | +4.1% | +3.0% | -3.4% | -0.1% | -3.4% | -1.4% | -0.0% |

Table 3: **Ablation Study of Target Inner-geometry Learning.** $\mathcal{L}_{\mathrm{depth}}^{R}$ and $\mathcal{L}_{\mathrm{bev}}$ denote the losses of inner-depth supervision and inner-feature BEV distillation, respectively.

| $\mathcal{L}_{\mathrm{depth}}^{R}$ | $\mathcal{L}_{\mathrm{bev}}$ | mAP | NDS |
|---|---|---|---|
| | | 0.329 | 0.431 |
| ✓ | | 0.339 | 0.440 |
| | ✓ | 0.359 | 0.454 |
| ✓ | ✓ | **0.366** | **0.461** |

Table 4: **Ablation Study of Inner-depth Supervision.** We compare different settings for relative depth value calculation and depth reference selection. * denotes our implementation.

| Setting | Depth Reference | mAP | NDS |
|---|---|---|---|
| BEVDepth* | - | 0.329 | 0.431 |
| All-to-Certain | 3D Center | 0.358 | 0.452 |
| | 2D Center | 0.358 | 0.452 |
| All-to-Adaptive | Highest Conf | 0.357 | 0.455 |
| | Smallest Error | **0.366** | **0.461** |

## 4.2 MAIN RESULTS

**On nuScenes Val and Test Set.** In Table 1, we compare our TiG-BEV with other 3D object detectors on nuScenes val set. As shown, our LiDAR-to-camera learning schemes respectively boost the three baseline models, BEVDet, BEVDet4D, and BEVDepth, by +2.9%, +1.4%, and +2.3% NDS. In Table 2, we report the performance on nuScenes test leaderboard. We can observe that our TiG-BEV outperforms all other multi-view BEV 3D detectors with 61.9% NDS and 53.2% mAP, and surpasses another LiDAR-to-camera learning method BEVDistill by +3.0% NDS and +4.1% mAP, achieving leading performance on this competitive benchmark. Besides, it shows that our method can also perform better with stronger backbones and larger resolutions.

**Visualization.** As visualized in Figure 7, we show results of BEVDepth before and after TiG-BEV. It is observed that more accurate results are obtained by our inner-geometry learning. Within the orange circles, false positives and ghosting objects can be reduced, and some 3D locations and orientations of the bounding boxes are also refined. More results are shown in Appendix B.2.

## 4.3 ABLATION STUDY

Here, we validate the effectiveness for each component and adopt BEVDepth as the student model.

**Inner-geometry Learning.** The individual effectiveness of the two main components can be examined by only equipping one of them. As shown in Table 3, if we only introduce the inner-depth supervision to the vanilla baseline, the mAP and NDS attains +1.0% and +0.9% gains. Instead, when we only use the inner-feature BEV distillation, the mAP and NDS are improved by +3.0% +2.3%. In addition, the combination of both components achieves better +3.7% mAP and +2.7% NDS, demonstrating the two proposed objectives can collaborate for better performance.

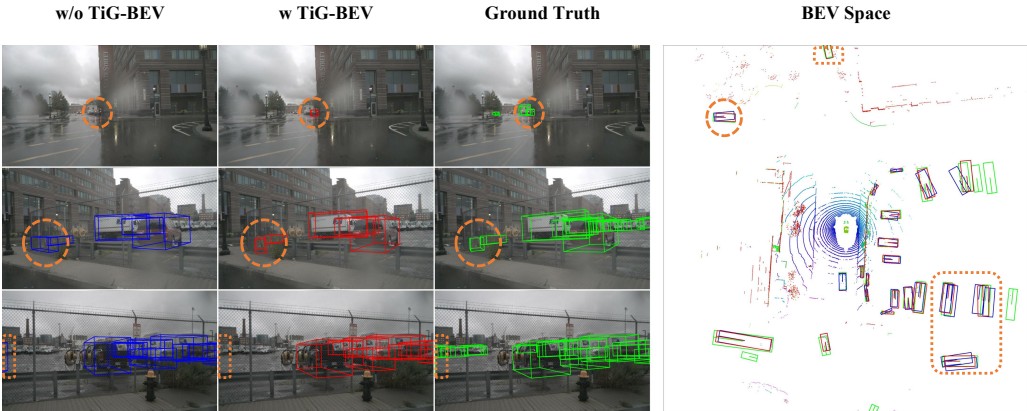

Figure 7: **Visualization of Detection Results**. From left to right, we show the results before and after the TiG-BEV learning schemes, ground-truth annotations, and the overall BEV-space results.

Table 5: **Ablation Study of Inner-feature BEV Distillation.** $\mathcal{L}_{bev}^{IC}$ and $\mathcal{L}_{bev}^{IK}$ denote the losses of inter-channel and inter-keypoint distillation, respectively.

| $\mathcal{L}_{depth}^{R}$ | $\mathcal{L}_{bev}^{IC}$ | $\mathcal{L}_{bev}^{IK}$ | mAP | NDS |
|---|---|---|---|---|
| | | | 0.339 | 0.440 |
| ✓ | ✓ | | 0.342 | 0.444 |
| | | ✓ | 0.358 | 0.452 |
| | ✓ | ✓ | **0.366** | **0.461** |

Table 6: **Target Depth Prediction Comparison.** * denotes our implementation.

| Metric | BEVDepth* | + TiG-BEV |
|---|---|---|
| SILog↓ | 29.306 | **15.897** |
| Abs Rel↓ | 0.217 | **0.091** |
| Sq Rel↓ | 1.103 | **0.232** |
| log10↓ | 0.079 | **0.038** |
| RMSE↓ | 3.167 | **2.077** |

**Inner-depth Supervision.** To calculate the relative depth values within foreground targets, we compare two paradigms concerning the relationships among different inner points, 1) *All-to-Certain* calculates the relative depth from all sampled points to a certain reference point, such as the projected center of 3D bounding box or the center of 2D bounding box. 2) *All-to-Adaptive* dynamically selects the reference pixel with the highest confidence across all depth bins or the smallest depth error to the ground truth (Ours). As shown in Table 4, our *All-to-Adaptive with smallest depth errors* obtains the best improvement, which indicates the dynamic depth reference point can flexibly adapt to different targets for inner-geometry learning.

**Inner-feature BEV Distillation.** Our TiG-BEV explores the BEV feature distillation from two perspectives, inter-channel and inter-keypoint. We equip the baseline with one of them at a time in Table 5. As shown, both distillation methods contribute to the final performance, respectively boosting the NDS by +1.3% and +2.1%. This well illustrates the importance of learning inner-geometry semantics within different foreground targets in BEV space. Further combining them two can benefit the performance by +3.7% and +3.0% for mAP and NDS. A more comprehensive comparison of distillation methods are shown in Table 7 of Appendix B.2.

**Target Depth Estimation.** As shown in Table 6, all metrics show that the depth prediction of target becomes more accurate with TiG-BEV, which is consistent with the visualization of predicted depth maps (Fig. 8) in Appendix B.2 and facilitates target positioning in BEV space.

## 5 CONCLUSION

In this paper, we propose a novel target inner-geometry learning framework that enables the camera-based detector to inherit the effective foreground geometric semantics from the LiDAR modality. We introduce inner-depth supervision and inner-feature distillation in BEV space, respectively for learning better low-level structures and high-level semantics. Extensive experiments are implemented to illustrate the significance of TiG-BEV for multi-view BEV 3D object detection. In future work, our focus will be on exploring a multi-modal learning strategy that enhances both camera and LiDAR modalities, aiming for a unified real-world perception system.

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

## A  MORE RELATED WORKS

**Depth Estimation.**  Depth estimation is a classical problem in computer vision. These method can be divided into single-view depth estimation and multi-view depth estimation. Single-view depth estimation is either regarded as a regression problem of a dense depth map or a classification problem of the depth distribution. (Bhat et al., 2021; Fu et al., 2018; Eigen et al., 2014; Poggi et al., 2020; Ranftl et al., 2021) generally build an encoder-decoder architecture to regress the depth map from contextual features. Multi-view depth estimation methods usually construct a cost volume to regress disparities based on photometric consistency (Wei et al., 2022; Guizilini et al., 2022; Zhang et al., 2019; Shen et al., 2021; Peng et al., 2022; Zhang et al., 2022d). For 3D object detection, previous methods (Park et al., 2021; Reading et al., 2021; Hong et al., 2022) also introduce additional networks for depth estimation to improve the localization accuracy in 3D space. Notably, MonoDETR (Zhang et al., 2022a;b) proposes to only predict the foreground depth maps instead of the dense depth values, but cannot leverage the advanced geometries provided by LiDAR modality. Different from them, our TiG-BEV conducts inner-depth supervision that captures local sptial structures of different foreground targets.

**Knowledge Distillation**  has shown very promising ability in transferring learned representation from the larger model (teacher) to the smaller one (student). Prior works (Zagoruyko & Komodakis, 2016; Huang & Wang, 2017; Liu et al., 2021a; Tung & Mori, 2019a) are proposed to help the student network learn the structural representation for better generalization ability. Such teacher-student paradigms have also been extended to other vision tasks, including action recognition (Cui et al., 2020), video caption (Pan et al., 2020), 3D representation learning (Fu et al., 2022; Zhang et al., 2022c; Liu et al., 2021b; Sautier et al., 2022), object detection (Dai et al., 2021; Chen et al., 2017; Zhou et al., 2023) and semantic segmentation (Hou et al., 2022; Wang et al., 2020). However, only a few of works consider the multi-modal setting between different sensor sources. For 3D representation learning, there are some interesting approaches. I2P-MAE (Zhang et al., 2022c) leverages Masked Autoencoders to distill 2D pre-trained knowledge into 3D transformers. BEV-LGKD Li et al. (2022a) generates the foreground mask and view-dependent mask for better localization. BEVDistill (Chen et al., 2022b) transfer knowledge from LiDAR feature to the cam feature by dense feature distillation and sparse instance distillation. UniDistill (Zhou et al., 2023) focuses on transferring knowledge from multi-modality detectors to single-modality detectors in a universal manner. X$^3$KD (Klingner et al., 2023) is a knowledge distillation framework for multi-camera 3D object detection, leveraging cross-modal and cross-task information by distilling knowledge from LiDAR-based detectors and instance segmentation teachers. DistillBEV (Wang et al., 2023) involves feature imitation and attention imitation losses across multiple scales, enhancing feature alignment between a LiDAR-based teacher and a multi-camera BEV-based student detector. BEVSimDet (Zhao et al., 2023) proposes a simulated multi-modal student to simulate multi-modal features with image-only input. VCD (Huang et al., 2023) presents some useful designs for temporal fusion and introduce a fine-grained trajectory-based distillation module. FD3D (Zeng et al., 2023) uses queries for masked feature generation and then intensify feature representation for refined distillation.

Our TiG-BEV also follows such teacher-student paradigm and effectively distills knowledge from the LiDAR modality into the camera modality. By modeling the relative relationships inside foreground objects in a novel way, the performances of camera-only detectors are further enhanced.

**Relationship Supervision.**  Some related works have investigated channel-wise and pixel-wise relationship supervision in various domains. (Gatys et al., 2016) studied pixel-wise relationships in image style transfer and found that matching higher layer style representations preserves local image structures at a larger scale, resulting in smoother visuals. (Tung & Mori, 2019b) proposed similarity-preserving knowledge distillation, guiding the student network towards teacher network's activation correlations. If two inputs produce similar activations in the teacher network, the student network should be guided towards a similar configuration. (Hou & Zheng, 2021) introduced a channel-wise relationship preserving loss for visualizing adapted knowledge in domain transfer. They claimed that channel-wise relationships remain effective after global pooling, unlike pixel-wise relationships, which can be overshadowed pre-classifier. Our TiG-BEV has also been inspired by these works and explored the designs of learning the internal relationships of foreground objects.

Table 7: **Comparison with BEVDistill (Chen et al., 2022b).** † and * denote the implementation of BEVDistill and ours, respectively. We present the performance improvement of the learning methods correspondingly to their implemented baselines.

| Method | mAP↑ | NDS↑ |
|---|---|---|
| BEVDepth† | 0.311 | 0.432 |
| + BEVDistill | 0.332 (+2.1%) | 0.454 (+2.2%) |
| BEVDepth* | 0.329 | 0.431 |
| + Naive Distill | 0.338 (+0.9%) | 0.434 (+0.3%) |
| **+ Inner-feature Distill** | **0.359 (+3.0%)** | **0.454 (+2.3%)** |
| **+ TiG-BEV** | **0.366 (+3.7%)** | **0.461 (+3.0%)** |

Table 8: **Comparison with Concurrent Works.** We present the performance improvement of some concurrent works, UniDistill (Zhou et al., 2023), X$^3$KD (Klingner et al., 2023), DistillBEV (Wang et al., 2023), correspondingly to their implemented baselines which uniformly use ResNet-50 as backbone with the image resolution of $256 \times 704$.

| Student | mAP↑ | NDS↑ | Method | mAP↑ | NDS↑ | Venue |
|---|---|---|---|---|---|---|
| BEVDet | 0.203 | 0.331 | UniDistill | 0.260(+5.7%) | 0.373(+4.2%) | CVPR 2023 |
| BEVDet | 0.305 | 0.378 | DistillBEV | 0.327(+2.2%) | 0.407(+2.9%) | ICCV 2023 |
| BEVDet | 0.298 | 0.379 | TiG-BEV | 0.331(+3.3%) | 0.411(+3.2%) | Ours |
| BEVDet4D | 0.328 | 0.459 | DistillBEV | 0.363(+3.5%) | 0.484(+2.5%) | ICCV 2023 |
| BEVDet4D | 0.322 | 0.451 | TiG-BEV | 0.356(+3.4%) | 0.477(+2.6%) | Ours |
| BEVDepth | 0.359 | 0.472 | X$^3$KD$_{modal}$ | 0.368(+0.9%) | 0.494(+2.2%) | CVPR 2023 |
| BEVDepth | 0.364 | 0.484 | DistillBEV | 0.389(+2.5%) | 0.498(+1.4%) | ICCV 2023 |
| BEVDepth | 0.357 | 0.481 | TiG-BEV | 0.383(+2.6%) | 0.498(+1.7%) | Ours |

# B MORE EXPERIMENTS

In this section, we further conduct a series of experiments to show the effectiveness of our approach.

## B.1 DATASET

NuScenes dataset (Caesar et al., 2020) provides synced data captured from a 32-beam LiDAR at 20Hz and six cameras covering 360-degree horizontally at 12Hz. We adopt the official evaluation toolbox provided by nuScenes, which reports the nuScenes Detection Score (NDS) and mean Average Precision (mAP), along with mean Average Translation Error (mATE), mean Average Scale Error (mASE), mean Average Orientation Error (mAOE), mean Average Velocity Error (mAVE), and mean Average Attribute Error (mAAE).

## B.2 MAIN RESULTS

**Comparison with BEVDistill (Chen et al., 2022b).** In Table 7, we compare our TiG-BEV with another LiDAR-to-camera learning method BEVDistill in the same setting. As shown, on top of a better baseline model, our approach can achieve higher performance boost for both mAP and NDS. Besides, we compare with the naive distillation that directly applies MSE loss to the entire BEV features between camera and LiDAR, where our inner-feature distillation performs better. These well demonstrate the superiority of target inner-geometry learning to foreground-guided dense distillation.

**Comparison with Concurrent Works.** As shown in Table 8, we further compare our TiG-BEV with other recent works in the same setting which also belong to LiDAR-to-camera learning meth-

Table 9: **Performance Comparison without CBGS (Zhu et al., 2019).** For all methods, we adopt ResNet-101 as the 2D backbone and $512 \times 1408$ as the image resolution. * denotes our implementation.

| Method | mAP↑ | NDS↑ |
|---|---|---|
| BEVDet* | 0.272 | 0.297 |
| **+ TiG-BEV** | **0.375 (+10.3%)** | **0.388 (+9.1%)** |
| BEVDet4D* | 0.336 | 0.435 |
| **+ TiG-BEV** | **0.409 (+7.3%)** | **0.489 (+5.4%)** |
| BEVDepth* | 0.393 | 0.487 |
| **+ TiG-BEV** | **0.430 (+3.7%)** | **0.514 (+2.7%)** |

Table 10: **Performance Comparison on KITTI Val Set (Geiger et al., 2012).**

| Method | 3D AP | | |
|---|---|---|---|
| | Easy | Moderate | Hard |
| CMKD (Hong et al., 2022) | 23.53 | 16.33 | 14.44 |
| + TiG-BEV | 26.63 | 16.61 | 14.31 |

ods. As there is a lack of consistency in the baselines reproduced by everyone, we have also included the baseline in the table for comparison. It should be noted that, the baseline performance of UniDistill is unexpectedly low, making it difficult to find an appropriate setting to compare with. Besides it, our proposed method has been demonstrated to be simple yet efficient and highly competitive, achieving comparable performance to the latest state-of-the-art methods.

**Without CBGS (Zhu et al., 2019) Strategy.** In Table 9, we present the results of TiG-BEV without the CBGS training strategy. Without the resampling of training data, the performance improvement of learning target inner-geometry becomes more notable, **+10.3%, +7.3%,** and **+3.7%** mAP for the three baselines, which indicates the superior LiDAR-to-camera knowledge transfer of our TiG-BEV.

**Performance on KITTI Val Set (Geiger et al., 2012).** In addition, we also conduct an experiment on KITTI validation set which is one of the most popular datasets for monocular 3D object detection. The main baseline is CMKD (Hong et al., 2022), which is a state-of-the-art method on KITTI dataset and use a classical distillation method to transfer knowledge of teacher model. We add constraints on knowledge distillation of target objects with our TiG-BEV for further optimisation and achieve a better performance than the baseline.

**Visualization.** More visualization results of BEVDepth before and after our TiG-BEV are shown in Fig. 9. With the help of our inner-geometry learning, the detection of false positives and ghost objects can be reduced, and more missed objects have been detected. The most obvious and common improvement is that locations and orientations of the bounding boxes are further refined and they are more consistent with the ground-truth boxes within orange marks. What's more, we visualize our depth prediction with and without the inner-depth supervision in Figure 8, which effectively refines the contours and edges of foreground objects. And the conclusion implied in the visualization of predicted depth maps is consistent with Table 6.

## B.3 ABLATION STUDY

**2D Backbones and Temporal Information.** We further explore the influence of 2D backbones and temporal information to our TiG-BEV in Table 11. We observe that our TiG-BEV brings significant performance improvement consistent over different 2D backbones. Also, our target inner-

**Before Inner-Depth**    **After Inner-Depth**

Figure 8: **Visualization of Predicted Depth Maps,** which are before and after the inner-depth supervision, respectively.

Table 11: **Ablation Study of 2D Backbones and Temporal Information.** CenterPoint (Yin et al., 2021) and BEVDepth (Li et al., 2022b) are adopted as the teacher and student models, respectively.

| Backbone | Resolution | Multi-frame | Method | mAP | NDS |
|---|---|---|---|---|---|
| VoxelNet | - | ✓ | Teacher | 0.564 | 0.646 |
| ResNet-18 | $256 \times 704$ | ✓ | Student
+ TiG-BEV | 0.285
**0.323 (+3.8%)** | 0.405
**0.430 (+2.5%)** |
| | | | Student
+ TiG-BEV | 0.260
**0.294 (+3.4%)** | 0.295
**0.335 (+4.0%)** |
| ResNet-50 | $256 \times 704$ | ✓ | Student
+ TiG-BEV | 0.329
**0.366 (+3.7%)** | 0.431
**0.461 (+3.0%)** |
| | | | Student
+ TiG-BEV | 0.298
**0.338 (+4.0%)** | 0.328
**0.375 (+4.7%)** |
| ResNet-101 | $512 \times 1408$ | ✓ | Student
+ TiG-BEV | 0.393
**0.430 (+3.7%)** | 0.487
**0.514 (+2.7%)** |
| | | | Student
+ TiG-BEV | 0.345
**0.403 (+5.8%)** | 0.366
**0.416 (+5.0%)** |

geometry learning schemes can provide positive effect for both single-frame and multi-frame settings. The improvement of mAP ranges from **+3.4%** to **+5.8%** and the improvement of NDS ranges from **+2.5%** to **+5.0%**.

**Performance of Small Object Detection.** In Table 12, we explored the impact of each component of our method on small object detection. We consider pedestrian, motorcycle, bicycle, traffic cone and barrier, these five classes in NuScenes to be relatively small objects, and we calculated their detection results in terms of mAP on the validation set. This quantitative analysis further confirms that our method is also effective in detecting small objects and provides gains in performance with each component.

**Performace of Different Distance Ranges.** The detection of distant objects remains a long-standing challenge due to the sparsity of LiDAR points and the inaccuracy of depth estimation. Here we define 0 to 30 meters as close detection, and 30 to 60 meters as long-distance detection. As can be seen from the Table 13, our method can greatly improves the performance of close detection, and it also works even in the face of long-distance object detection with each designed module. In the future work, we will continue to pay attention to and strive to solve the problem of long-distance object detection.

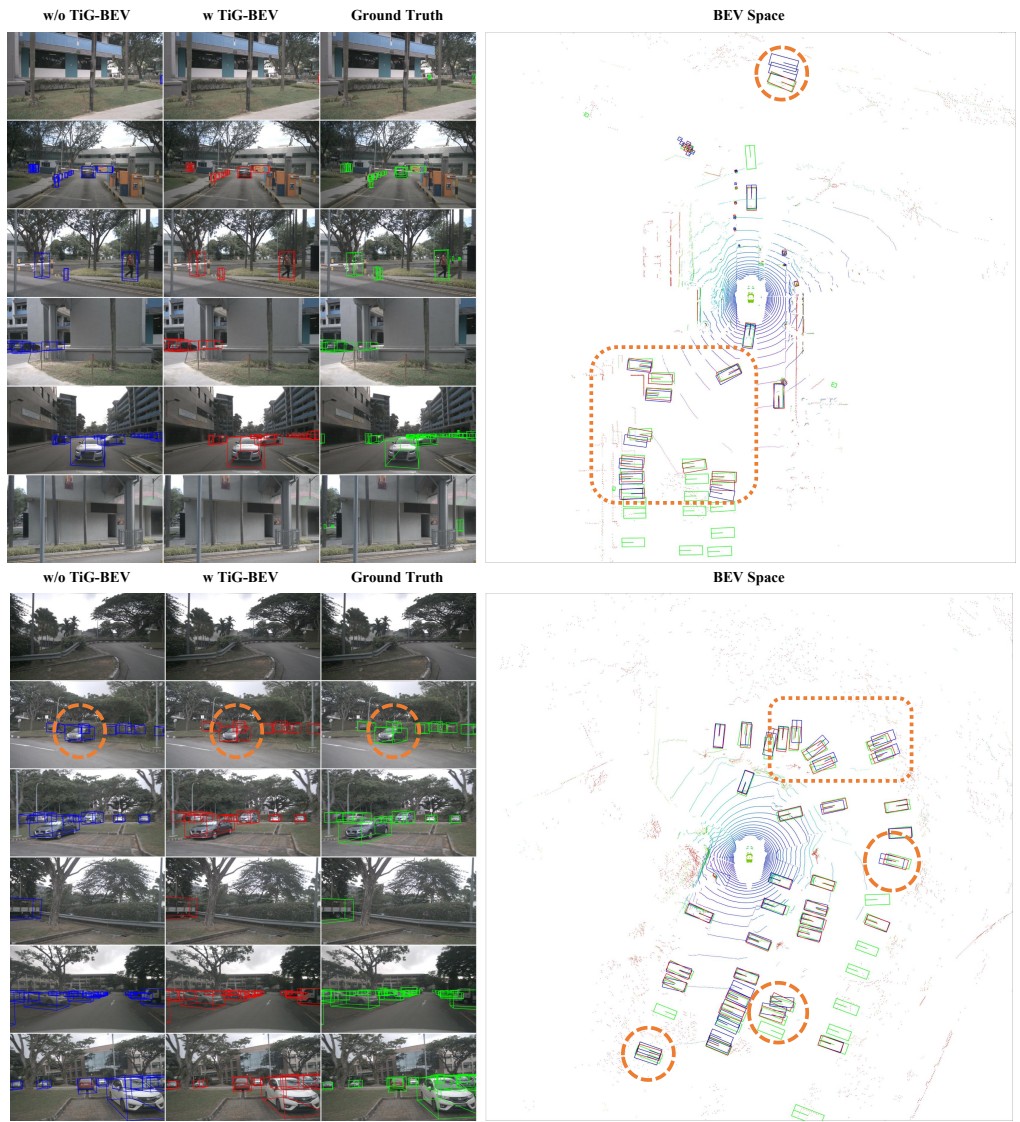

Figure 9: **Visualization of Detection Results**. From left to right, we show the 3D object detection before and after the TiG-BEV learning schemes, ground-truth annotations, along with the overall BEV-space results.

Table 12: **Ablation Study of Small Objects Detection Performance.** We use BEVDet4D as baseline and ResNet-101 as backbone with the image resolution of $512 \times 1408$.

| $\mathcal{L}_{\mathrm{depth}}^{A}$ | $\mathcal{L}_{\mathrm{depth}}^{R}$ | $\mathcal{L}_{\mathrm{bev}}$ | mAP↑ (%) | | | | |
| :---: | :---: | :---: | :---: | :---: | :---: | :---: | :---: |
| | | | pedestrian | motorcycle | bicycle | traffic_cone | barrier |
| | | | 43.3 | 34.5 | 32.4 | 54.8 | 55.2 |
| ✓ | | | 44.6 | 35.6 | 32.8 | 56.3 | 55.9 |
| ✓ | ✓ | | 45.4 | 37.3 | 33.7 | 57.8 | 58.8 |
| ✓ | ✓ | ✓ | **45.7** | **38.8** | **37.5** | **58.9** | **57.2** |

Table 13: **Ablation Study for Objects Detection Performance of Different Distance Ranges.** We use BEVDet4D as baseline and ResNet-101 as backbone with the image resolution of $512 \times 1408$.

| Distance (m) | $\mathcal{L}_{\text{depth}}^{A}$ | $\mathcal{L}_{\text{depth}}^{R}$ | $\mathcal{L}_{\text{bev}}$ | mAP↑ | NDS↑ |
|---|---|---|---|---|---|
| [0,30) | | | | 44.2 | 53.7 |
| [0,30) | ✓ | | | 46.6 | 55.6 |
| [0,30) | ✓ | ✓ | | 47.4 | 55.8 |
| **[0,30)** | ✓ | ✓ | ✓ | **48.6** | **56.8** |
| [30,60) | | | | 10.2 | 28.7 |
| [30,60) | ✓ | | | 10.3 | 28.5 |
| [30,60) | ✓ | ✓ | | 11.7 | 30.1 |
| **[30,60)** | ✓ | ✓ | ✓ | **12.6** | **30.2** |

## B.4 DISCUSSION

**Improvements of Inner-depth Supervision vs. Inner-feature Distillation.** Our inner-depth provides fine-grained depth cues for object-level geometry understanding, which is low-level information and implicitly benefits the mAP accuracy. The inner-feature directly supervises high-level semantics of BEV features, explicitly affecting the detection and recognition performance. They are complementary and can collaborate for better results.

**Limitations and Future Works of Inner-depth Supervision.** One limitation that will inevitably be involved in depth-related explicit supervision is the ground truth from LiDAR. Due to the sparsity of LiDAR, both absolute depth supervision and relative depth supervision will be affected to some extent, though the performance of object detection will be better with inner-depth supervision under the same conditions. For sparse point clouds, a good way to alleviate it is through depth completion.

Another limitation is the occlusion problem of the target. Due to visual occlusion, the ground truth of the occluded object that we can obtain is limited, which will affect the learning of depth prediction and inner-depth learning. A good way to alleviate it is to consider multi-frame images for inner-depth supervision of the same object's interior. But a potential risk here is that if the intrinsic and extrinsic parameters are inaccurate, it will directly affect the coordinate system transformation between multiple frames, thereby affecting the ground truth of depth values.

