# OpenReview forum: "TiG-BEV: Multi-view BEV 3D Object Detection via Target Inner-Geometry Learning"
_ICLR.cc/2024/Conference — ICLR 2024 Conference Desk Rejected Submission_

### Official Review · Reviewer_kyMY · 2023-10-30

**Soundness:** 2 fair
**Presentation:** 2 fair
**Contribution:** 2 fair
**Rating:** 6
**Confidence:** 4

**Summary:**

TiG-BEV proposes an internal geometric learning scheme for the target, which enhances the camera based BEV detector from both depth and BEV features by utilizing LiDAR mode. Firstly, an internal deep supervision module is introduced to learn the low-level relative depth relationships of each target, thereby enabling the camera detector to have a deeper understanding of the target level spatial structure. Secondly, an internal feature BEV distillation module was designed to mimic the high-level semantics of different key points within the foreground target. In order to reduce the domain difference between the two modes, distillation within the channel and between key points was used to model feature similarity.

**Strengths:**

The research methodology proposed in this paper involves two main components.
1. An inner-depth supervision module is introduced to learn the low-level relative depth relations within each object. This helps the camera-based detectors gain a deeper understanding of object-level spatial structures.
2. An inner-feature BEV distillation module is designed to imitate the high-level semantics of different keypoints within foreground targets using inter-channel and interkeypoint distillation.

**Weaknesses:**

1. The improvement over other methods is limited, as there have been numerous distillation methods proposed for camera-based detectors such as BEV-LGKD, DistillBEV, BEVSimDet, and X3KD. However, this method lacks clear advantages compared to previous approaches, and no extensive comparison with other methods has been provided.
2. The comparison is biased. Firstly, BEVDistill is specifically designed for BEVFormer, which only has one distillation module for BEVDepth. Secondly, the experiments conducted on the test set were augmented, which does not provide a fair comparison. Thirdly, I noticed that the baseline model achieved similar scores to BEVDistill in Table 2, but the official code does not provide the checkpoint for that. Therefore, I strongly recommend conducting comparisons using the Res-50 model to demonstrate improvements over previous methods such as BEV-LGKD, DistillBEV, BEVSimDet, and X3KD.

**Questions:**

1. The paper should report results based on long-term temporal methods, such as solofusion, to demonstrate the effectiveness of the proposed method.
2. The paper should provide a more comprehensive and fair comparison with previous methods, specifically in the ResNet-50 (R50) framework, to showcase the improvement of the proposed method over existing approaches such as BEV-LGKD, DistillBEV, BEVSimDet, and X3KD. This will provide a clearer understanding of the performance gains achieved by the proposed method.

---

> ### Author Response · Authors · 2023-11-21
> **Response to Reviewer kyMY**
>
> Thank you for your time and effort.
>
> ### **Q1. Compare with other related works in the R50 framework.**
>
> Thank you for your suggestion. We have shown the comparison in **General Response to All Reviewers**. We are sure that our comparison is fair. According to the Reviewer Policy, we have reported the results of several methods using the LiDAR->Camera mode on ResNet50 and published in top-tier conferences. As there is a lack of consistency in the baselines reproduced by everyone, we have also included the baseline in the table for comparison. It is evident that our method performs competitively. We hope it can provide a clearer understanding.
>
> ### **Q2. The comparison is biased?**
>
> We apologize for any confusion and misunderstanding that we may have caused you. We need to state that we place great emphasis on comparing fairly and we have enough confidence to ensure that our comparison is fair.
>
> - Firstly, BEVDistill also used two distillation modules (BEV Feature Level and Instance Response Level) proposed in the paper for the BEVDepth experiment, which is referred to in Appendix B.3 in BEVDistill. These two modules are silimar to distillation methods in 2D object detection and are proved to be generalizable. The author adapted instance-level distillation on two different frameworks (bevdepth and bevformer), while bev feature level distillation is generalizable, like our inner-feature distillation. Moreover, as shown in Table 7 in our paper, we also list the baseline results and show the improvement respectively to make a fair comparison.
>
> - Secondly, both BEVDistill and our method used the same and common data augmentation for training, such as image flipping, scaling, etc., and neither of us used other augmentation methods such as TTA for test set. Therefore, there is no unfair comparison here.
>
> - Thirdly, we are a little confused about your comment. We are not quite sure which paper's 'Table 2' you are referring to (BEVDepth/BEVDistill/TiG-BEV?). The Table 2 in our paper shows the NuScenes Test Leaderboard result, all participants do not need to, and are not allowed to provide checkpoints for it. While the results we provide for the baseline are consistent with the results reported by BEVDistill, because our implementation of the baseline method is also completely consistent. We all use the BEVDet_v1.0 codebase to reproduce the result of the BEVDepth Baseline. What's more, the config of the baseline we used was given by the author of BEVDistill, we had ever asked for it. So our baselines for test leaderboard are the same and we also found that our results reproduced by the same config were also the same. Therefore, from Table 2, we can see the superiority of our method clearly and it's also completely fair.
>
> ### **Q3. Report results based on long-term temporal methods.**
>
> Thanks for your suggestion. We did not specifically design for long-term setting before. In order to show results based on long-term setting, we trained a LiDAR Teacher with BEVDet_v2.0 codebase [1]. Due to resource limitations, our teacher model appears to be not powerful enough (mAP:49.7/ NDS:52.3), but it still can be effective. Without any tuning, we were still able to further improve the BEVDet4D-Depth on the long-term setting (8 + 1 frames) as shown in the table.
>
>
> |Method|w/ TiG-BEV |mAP $\uparrow$|NDS $\uparrow$|
> |--- | --- | --- | --- |
> |BEVDepth|   |39.4|51.5|
> |BEVDepth| $\checkmark$  |41.3|52.5|
>
>  #### Note: For the table, we use ResNet50 as the backbone, the resolution is 256 $\times$ 704.
>
>
> &nbsp;
> &nbsp;
> &nbsp;
> &nbsp;
>
> Overall, thank you for bringing your concerns to our attention. We hope that our response has effectively addressed them and cleared up any misunderstandings.
>
> ### Reference:
>
> [1] https://github.com/HuangJunJie2017/BEVDet/tree/dev2.0

---

> ### Comment · Reviewer_kyMY · 2023-11-22
> **Rebuttal Feedback**
>
> The reviewer expresses gratitude to the authors for their feedback; nevertheless, there is a lingering issue as some of his concerns remain unaddressed.
>
> Q1. The progress in points is somewhat limited compared to previous approaches such as DistillBEV. He maintains that this paper presents only a narrow scope of new insights, as emphasized by Reviewer oss3. Regarding the feature distillation component, its characteristics did not appear markedly distinct from those of DistillBEV. Instead, it seemed predominantly foregrounded, albeit in a divergent manner. Besides, through ablation experiments, it has been ascertained that the depth distillation module yields only marginal improvements.
>
> Q2. The reviewer acknowledges the inclusion of new references to established works. However, there are still some works not included as follows.
>
> [1]. BEVSimDet: Simulated Multi-modal Distillation in Bird's-Eye View for Multi-view 3D Object Detection.
>
> [2]. Leveraging Vision-Centric Multi-Modal Expertise for 3D Object Detection.
>
> [3]. Distilling Focal Knowledge From Imperfect Expert for 3D Object Detection.
>
> [4]. BEV-LGKD: A Unified LiDAR-Guided Knowledge Distillation Framework for BEV 3D Object Detection.

---

> > ### Author Response · Authors · 2023-11-22
> > **Response to Feedback by Reviewer kyMY**
> >
> > Thank you very much for your quick reply. We hope our further response can address your concerns.
> >
> > **Q1.**
> >
> > - Firstly, we would like to remind the reviewers that **DistillBEV is somewhat our concurrent work**, which is firstly released on arXiv on September 26th, 2023, within two months to ICLR submission deadline. The improvement we bring to the baseline is highly competitive, both on the validation set and the test set of nuScenes. Especially on the test set, where **we can bring even more improvement to a similar and strong baseline**. What's worth noting is that **we only used a LiDAR-based teacher**, while DistillBEV used a LiDAR-Camera Fusion teacher.
> >
> >     |Backbone |Student  | mAP $\uparrow$| NDS $\uparrow$|Method|mAP $\uparrow$| NDS $\uparrow$|Venue|
> >      |--- | --- | --- | --- | --- | --- | --- | --- |
> >      |Swin-B    | BEVDepth   | 48.9   | 59.0   | DistillBEV  | 52.5$_{(+3.6)}$   | 61.2$_{(+2.2)}$   | ICCV 2023
> >      |**ConvNeXt-B**|**BEVDepth**|**49.1**|**58.9**|**TiG-BEV**  |**53.2$_{(+4.1)}$**|**61.9$_{(+3.0)}$**|**Ours**
> >
> >      #### Note: In this table, we show the performance evaluated in the test set. Our proposed method is implemented by BEVDet_v1.0 and the image resolution is 640 $\times$ 1600. We don't use TTA and extra data.
> >
> >
> > - Secondly, as we shown in **General Response to All Reviewers**, we discuss about the main differences among recent works.
> >   - Regarding DistillBEV, it's feature distillation component is specially designed for strict feature alignment between different modalities and leverages multi-scale techniques in carefully selected layers. While our method **is different from feature alignment used in most of recent works** and looks into relationships inside the foreground objects to relieves the modality gap as illustrated in our teaser figure (Fig 1). Compared to DistillBEV, we not only consider the spatial information but also take semantic and depth information into consideration. They all serve our motivation to learn a better BEV representation by leveraging the internal connections of objects with the help of the teacher model.
> >   - From the perspectives of inner-depth and inner-feature, we enable the model to fully learn the knowledge of internal relationships contained in the teacher model, rather than simply imitating the teacher model's feature values. This is also a key contribution of our work, and extensive experiments have shown that it is effective and beneficial for further enhancing the performance of camera-based student detectors.
> >   - We pay attention to foreground objects because of their useful information contained in internal relationships while a large amount of background information is redundant in the BEV grid. In addition to that, **we also pay attention to background information to a certain extent**. For example, when doing absolute depth supervision, we supervise the entire depth map containing the background, which is beneficial to depth prediction. While inner-depth supervision is used to strengthen the connection between the depths within the foreground. When distilling the BEV feature map, in addition to sampling points within the foreground target, we also consider the background information at the foreground edge to obtain clearer boundaries and better distinguish the features between foreground and background.
> >
> >
> > - Thirdly, as we discussed in Appendix B.4, our inner-depth provides fine-grained depth cues for object-level geometry understanding, which is low-level information and implicitly benefits the mAP accuracy. While the inner-feature directly supervises high-level semantics of BEV features, explicitly affecting the detection and recognition performance. They are complementary and can collaborate for better results.
> >
> >
> > **Q2.** Thanks you for the additional references, we have taken note of these relevant works and introduced them in Appendix A in our latest manuscript. If there is anything that we have misunderstood, please kindly point it out to us. We would greatly appreciate it.

---

> ### Comment · Area_Chair_HYDj · 2023-12-01
>
> Hi Reviewer,
>
> This paper obtained the mixed opinion and I see your comments carefully. I wonder whether there are some misunderstandings in this paper. So could you give the updated comments given the author's rebuttal?
>
> It might be helpful for us to score this paper.
>
> Many thanks,
>
> AC

---

### Official Review · Reviewer_CVGt · 2023-10-31

**Soundness:** 3 good
**Presentation:** 3 good
**Contribution:** 2 fair
**Rating:** 3
**Confidence:** 4

**Summary:**

The authors propose a new method for multi-camera 3D object detection, where existing techniques for knowledge distillation from a pre-trained LiDAR-based detector to a camera-based detector are enhanced. The authors propose to extract a reference point in the image space as well as keypoints in bird’s eye view and focus the distillation from LiDAR-based models to camera-based models on these points. Evaluation on the nuScenes dataset shows that the method is able to outperform various previous baseline methods.

**Strengths:**

-	The introduction and related work provide a clear and concise motivation for the proposed method. Figures 1 and 2 give a nice qualitative insight into the proposed contributions.
-	The method description is clear and easy to follow. The mathematical description is overall quite precise and Figures 2-6 help a lot to better understand the method.
-	The authors show that their method can be combined with various baseline methods.
-	The ablation study verifies the effectiveness of the single method components.

**Weaknesses:**

Issues:

1.	Recently, there have been some relevant related works [a, b], which seem quite related to the method presented in this paper. I think it would be good to compare to these works in the SOTA comparison as well and discuss differences in a bit more detail.
[a] Klingner et al. “X3KD: Knowledge Distillation Across Modalities, Tasks and Stages for Multi-Camera 3D Object Detection,” CVPR 2023
[b] Zhou er al. “UniDistill: A Universal Cross-Modality Knowledge Distillation Framework for 3D Object Detection in Bird's-Eye View,” CVPR 2023

2.	I am wondering how generalizable the method is. It seems that especially the improvement in terms of learning a better depth distribution is quite constraint to methods making use of feature projection via depth maps. However, e.g., DETR-like methods for multi-camera 3DOD do not have this feature. It would be interesting to provide insights how the method could be extended to such methods.

3.	Due to the two mentioned issues I feel that the contribution and scope of this works appears a bit limited. If they can be addressed in the rebuttal, I am open to reconsider this point.

Minor comments and typos:

4.	Section 3.1: I think it would be good to have a better differentiation in terms of notation between the number of depth bins D and the depth map D. I also did not completely understand the upper index A in the depth loss.

5.	Section 3.1: I think it is a bit confusing to talk about 3d and 2d features if they have exactly the same shape. The same goes for 3D and 2D detection heads in Figure 4, which are probably exactly the same just for different modalities as input. Maybe it would help to rather talk about camera and LiDAR features?

6.	Figure 4: It would be nice to include the math notation also in the figure to be able to better connect the method description and the visualization of the method in the figure

7.	The text in Figure 5 and 6 is quite small. It would be nice to increase the font size a bit.

8.	Table 1 and 2: It would be good to include the latest works on multi-camera 3DOD in the SOTA comparison, e.g., [a,b] or to explain why they are excluded. As far as I saw, they also report on nuScenes, so shouldn’t they be comparable?

9.	It would be nice to verify the method on more than one dataset, e.g., the Waymo dataset.

10.	Table 2: It would be good to also report resolution and backbone in this comparison.

11.	There are a few (minor) typos remaining throughout the whole paper. It would be good to resolve them with another round of proofreading.

**Questions:**

-	It is interesting to see that using only a reference point for depth supervision is superior to using a dense depth map as supervision. I have not completely understood the motivation though. Could the authors provide a bit more insight, maybe also from their experience from qualitative results? Also, I was asking myself what happens if there is no ground truth available at the reference point since supervision usually comes from sparse LiDAR data?
-	Similarly, could the authors maybe explain why it is beneficial to uniformly sample key points from the bounding box in BEV space instead of supervising throughout the whole area of the bounding box?

---

> ### Author Response · Authors · 2023-11-21
> **Response to Reviewer CVGt - Part1**
>
> Thanks for your constructive and highly detailed comments that help us a lot.
>
> ### **Q1. Compare with other related works and discuss differences.**
>
> Thanks you very much for this suggestion, we make the comparison in **General Response to All Reviewers**, please refer to it for a more clear understanding. In the table, we list three relevant related works published in top-tier conferences and discuss our differences among these works.
>
> ### **Q2: How generalizable the method is?**
>
> It's an interesting topic. Your inquiry has provided valuable insights, motivating us to reevaluate the contributions of our method and explore potential synergies with other approaches.
>
> The framework of the DETR-like series leverages implicit depth information and position encoding. In this paradigm, learnable queries, preset on the BEV grid, are employed to seek relevant image features based on their spatial location in the BEV space. This approach contrasts with the LSS-like methodology, where explicit depth information supervises the process, projecting image features onto the BEV. It's noteworthy that our supervision includes the relative depth within foreground objects, constituting a form of explicit guidance for depth prediction.
>
> However, the DETR-like approach encounters challenges when querying image features due to occlusion and other phenomena in the image. This often leads to inconsistencies between the detected image features and their counterparts in the BEV space, resulting in occasional false detections. Addressing these issues with implicit depth information remains a challenge.
>
> In recent developments, we've observed innovative approaches, such as FB-BEV [1]. specifically, FB-BEV ultilizes BEVFormer with depth-aware module, like BEVDepth, integrates LSS-like module for explicit depth supervision to enhance DETR-like module. This method aims to harness the complementary strengths of both methods, yielding a more accurate BEV representation.
>
> Similarly, we recognize the potential for enhancing the DETR-like approaches by incorporating our method and our experimental validation of this concept has yielded promising results. Essentially, our method acts as a plug-and-play module that seamlessly integrates with and improves the DETR-like approach by utilizing auxiliary inner-depth information, akin to the approach outlined in FB-BEV.
>
>
> |Method|$\mathcal{L}^R_{\rm{depth}}$ | $\mathcal{L}_{\rm{bev}}$ |mAP $\uparrow$|NDS $\uparrow$|
> |--- | --- | --- | --- | --- |
> |FB-BEV|  |  |32.8|41.5|
> |FB-BEV| $\checkmark$|  |34.1|42.8|
> |FB-BEV| $\checkmark$| $\checkmark$|35.0|43.6|
>
>  #### Note: For the table, we use ResNet50 as the backbone, the resolution is 256 $\times$ 704. We don't use temporal information due to limited resources.
>
>  ### **Q3: Minor comments and typos.**
>
>
> We sincerely apologize for any writing issues in the previous version. We have diligently addressed all typos and continued fine-tuning the paper's writing to enhance its overall quality. Our team is actively engaged in preparing for the final submission, and we are committed to delivering a polished and refined manuscript. Thank you for your patience and guidance.
>
>
> > (1) The upper index A in the depth loss?
>
> The upper index A in the depth loss means "Absolute", we use absolute depth supervision like BEVDepth. The loss was named with the first letter A of **A**bsolute to correspond and distinguish it from the **R**elative inner-depth supervision loss we proposed.
>
> > (2~5) A bit confusing to talk about 3d and 2d features; Include the math notation also in the Fig 4; Increase the font size of Fig 5,6 a bit; Report resolution and backbone in Tab 2.
>
> Sorry for your confusion and thank you for your suggestion. We have made modifications according to it and updated in the latest manuscript.
>
> > (6) Include the latest works on multi-camera 3DOD in Tab 1,2.
>
> We have updated Table 2 in the latest manuscript. For the sake of fairness in comparison, we have included the detailed table in Appendix B.2 (Table 8), as we did in the **General Response to All Reviewers**.
>
> > (7) Verify the method on more than one dataset.
>
> As for other datasets, we have already shown the performance of our method on KITTI in Appendix B.2 (Table 10).
>
> If you have any other suggestions, we welcome and appreciate your continued feedback.

---

> > ### Author Response · Authors · 2023-11-21
> > **Response to Reviewer CVGt - Part2**
> >
> > ### **Q4: Only a reference point for depth supervision is superior to using a dense depth map as supervision?**
> >
> > We sincerely apologize for any confusion caused by our explanation. Allow us to provide further clarification on our approach.
> >
> > In our method, we employ a dual supervision strategy, combining dense depth map supervision with inner-depth supervision which is based on a reference point. According to our experimental observations, we have found that a more precise depth prediction is crucial for inner-depth supervision, as it facilitates a refined reference point.
> >
> > However, the supervision of each pixel for dense depth map through independent supervision may lead to inaccurate depth prediction in certain areas of the object, such as edges. If these areas are also used as reference points for inner-depth supervision, it may have a negative impact. Therefore, it is important to adaptively select a reference point with relatively accurate depth. The incorporation of inner-depth supervision aims to move beyond isolated point predictions and consider the relative depth relationships between objects. This nuanced approach enhances the quality of the BEV representation.
> >
> > ### **Q5: What happens if there is no ground truth available?**
> >
> > If there is no ground truth for a certain area, methods that only use depth information from sparse point clouds for supervision, such as BEVDepth, will lack depth supervision for this area. while since there are not too many of these areas in the overall depth map, their impact on the model's learning will not be significant.
> >
> > But with richer and more accurate depth information, such as a complete dense depth map, our approach can perform better. The reason why we utilize sparse LiDAR points for auxiliary depth information is for a fair comparison with BEVDepth.
> >
> > We appreciate your understanding and are committed to providing a clearer and more comprehensive description in the revised manuscript.
> >
> >
> > ### **Q6: Why it is beneficial to uniformly sample key points from the bounding box in BEV space instead of supervising throughout the whole area of the bounding box?**
> >
> >
> > Thank you for your insightful inquiry into the intricacies of our module design. The key objective of our method is to capture the relationships among points within an object, necessitating the calculation of relative relationships among these internal points. However, to ensure computational efficiency and address variations in bounding box sizes across different object categories and scenes, we opt for a uniform sampling approach in BEV space. This strategy involves selecting key points from the bounding box, preventing excessive computational overhead and accommodating diverse object sizes in different scenarios.
> >
> > Besides, if we were to directly do the point-to-point feature alignment of the whole area across various modalities (similar to BEVDistill), it would be susceptible to certain modality differences, as highlighted in our paper.
> >
> >
> > &nbsp;
> > &nbsp;
> > &nbsp;
> > &nbsp;
> >
> > Overall, we sincerely appreciate the effort and diligence you've invested in reviewing our paper. Your recognition and detailed feedback have been invaluable to us. We hope that our response effectively addresses your concerns.
> > Should you have any further questions or suggestions, we are more than willing to engage in further discussion.
> >
> > ### Reference:
> >
> > [1] *Li Z, Yu Z, Wang W, et al. Fb-bev: Bev representation from forward-backward view transformations[C]//Proceedings of the IEEE/CVF International Conference on Computer Vision. 2023: 6919-6928.*

---

> > > ### Author Response · Authors · 2023-11-22
> > > **Sincere Request for Further Discussions**
> > >
> > > Dear Reviewer CVGt,
> > >
> > > Thanks again for your great efforts and constructive advice in reviewing this paper! With the discussion period drawing to a close, we expect your feedback and thoughts on our reply. We put a significant effort into our response, with several new experiments and discussions. We sincerely hope you can consider our reply in your assessment. We look forward to hearing from you, and we can further address unclear explanations and remaining concerns if any.
> > >
> > > Regards,
> > >
> > > Authors

---

### Official Review · Reviewer_oss3 · 2023-11-01

**Soundness:** 3 good
**Presentation:** 3 good
**Contribution:** 2 fair
**Rating:** 5
**Confidence:** 4

**Summary:**

this paper investigates the problem of BEV 3D object detection from multi-view RGB images. using the LiDAR as teacher supervision, this manuscript introduces a relative depth supervision and relationship matching for knowledge distillation. the effectiveness of the proposed method is verified on nuScenes dataset.

**Strengths:**

+ relatively easy to read
+ good results
+ ablation and variant study

**Weaknesses:**

the major issue that the reviewer has with this current manuscript is that it does not introduce significant new knowledge to the readers.
- the proposed 'inner-depth supervision' is a object-normalized version of the absolute depth from BEVDepth. yes, normalizing the depth within the objects can make it easier to learn, where the absolute depth (normalized to 1) might range from 0.1 to 0.15, and the relative depth can be 0 and 1 correspondingly. it is great that the ablation shows improvement, but this is to be expected and well-recognised, and feels more like a trick.
- the channel-wise and pixel-wise relationship supervision in Section 3.3 has been investigated by previous works [r1,r2,r3] on multiple tasks, so their capability in the BEV detection task (Table 3 and Table 5) is also somewhat expected. also, please refer to related works on 'relationship supervision', which could greatly benefit readers new to this field.

overall, the reviewer does not feel entirely confident in recommending this manuscript at its current state.



[r1]. Gatys, Leon A., Alexander S. Ecker, and Matthias Bethge. "Image style transfer using convolutional neural networks." In Proceedings of the IEEE conference on computer vision and pattern recognition, pp. 2414-2423. 2016.

[r2]. Tung, Frederick, and Greg Mori. "Similarity-preserving knowledge distillation." In Proceedings of the IEEE/CVF international conference on computer vision, pp. 1365-1374. 2019.

[r3]. Hou, Yunzhong, and Liang Zheng. "Visualizing adapted knowledge in domain transfer." In Proceedings of the IEEE/CVF conference on computer vision and pattern recognition, pp. 13824-13833. 2021.

**Questions:**

see weakness

---

> ### Author Response · Authors · 2023-11-21
> **Response to Reviewer oss3**
>
> Thanks for your recognition and valuable feedback.
>
> ### **Q1. The proposed 'inner-depth supervision' is a object-normalized version of the absolute depth from BEVDepth?**
>
> We are very sorry for the confusion. We propose the inner-depth supervision module, which is mainly motivated by focusing on the internal relationships of foreground objects. It's not a trick. The introduction of relative depth is different from normalization which amplifies the differences between similar depth clusters in a certain area. It does not involve any normalization operations. Insteadly, it establishes a connection between the depths of the internal points of foreground objects, rather than predicting the depth of each point in isolation.
>
> Absolute depth supervision ignores the implicit relationships between points, while our inner-depth supervision aggregates the depths of points belonging to the same object, allowing the model to learn the relationships between internal points of objects and project object features more accurately into the BEV space.
>
> A more intuitive visualization after inner-depth supervision can be seen in Appendix B.2 (Fig 8), and even the depth prediction of the edge has become clearer.
>
> ### **Q2. The channel-wise and pixel-wise relationship supervision has been investigated by previous works?**
>
> Thanks for the additional references, we have added an extra paragraph of these related works in Appendix A according to your kind advice.
>
> We agree that some related work has explored the relationship supervision between features in the fields of knowledge distillation and style transfer. These involve knowledge distillation between teachers and students of different scales or knowledge transfer between different domains, but few works have been done on different modalities of teachers and students, where one feature is learned from 2D data and the other from 3D data. Models under these two different modalities not only differ in model parameters and feature domains, but also in data format and source, which poses some difficulties that are not present in normal relationship learning.
>
> Besides, in normal knowledge distillation, direct feature alignment can be a good auxiliary method, but in our setting, differences between modalities can be harmful. Therefore, our work has conducted in-depth exploration on this issue. From the experimental results, it is evident that our method is feasible and can significantly outperform direct feature alignment, which not only meets expectations but also makes sense.
>
> In terms of specific design, we have made detailed designs through extensive trials. We did not learn the relationship of the entire feature map, because a large amount of background noise can cause interference and heavy computational burden. Through exploration, we applied relationship supervision to foreground objects which is more efficient and can bring better improvement. Moreover, we used uniform sampling points to learn the relationship within foreground objects to save computational costs. In addition, we also found that learning both spatial correlation and channel-level correlation can complement each other. Finally, we found that learning feature correlation implicitly depends on previous depth prediction. The inner-depth supervision we used better constrained the feature projection at the foreground level, which further assisted in the learning of inner-feature. The combination of these two modules can achieve better results. Thus, our paper proposes a method that integrates and complements each other.

---

> > ### Author Response · Authors · 2023-11-21
> > **Response to Reviewer oss3 - part2**
> >
> > Here is the additional paragraph we added in the latest manuscript:
> >
> > **Relationship Supervision.** Some related works have investigated channel-wise and pixel-wise relationship supervision in various domains. (Gatys et al., 2016) studied pixel-wise relationships in image style transfer and found that matching higher layer style representations preserves local image structures at a larger scale, resulting in smoother visuals. (Tung & Mori, 2019b) proposed similaritypreserving knowledge distillation, guiding the student network towards teacher network’s activation correlations. If two inputs produce similar activations in the teacher network, the student network should be guided towards a similar configuration. (Hou & Zheng, 2021) introduced a channel-wise relationship preserving loss for visualizing adapted knowledge in domain transfer. They claimed that channel-wise relationships remain effective after global pooling. Our TiG-BEV has also been inspired by these works and explored the designs of learning the internal relationships of foreground objects.
> >
> > If there is anything that we have misunderstood, please kindly point it out to us. We would greatly appreciate it.

---

> > ### Comment · Reviewer_oss3 · 2023-11-22
> > **rebuttal feedback**
> >
> > the reviewer thanks the authors for their feedback. however, his concerns are not fully addressed.
> >
> > Q1. regarding the 'inner-depth supervision', the updated equations 3 and 4 verify the point raised by the reviewer that the 'inner-depth' is an object-normalized version of the absolute depth, albeit there is only subtraction of 'mean' (see the subtraction of the reference depth in Eq. 3) and no division by 'std'. the description "amplifies the differences between similar depth clusters in a certain area" also feels like per-object 'normalization', where the subtraction of a large reference value helps the network to focus on the nuances. overall, this per-object 'normalization' with subtraction still feels like a trick to the reviewer.
> >
> > the reviewer has a different suspicion of the working mechanism underneath. enforcing the *softmax averaged* depth in Eq. 1 helps the categorical depth $D_i$ to be *softer* and provides more granularity (continuous) than splitting the depth into D bins (discrete). this in turn helps the BEV feature representation in LSS or BEVDet, which is also *softmax averaged* using the now potentially 'softer' or better categorical depth $D_i$. would it be possible to run ablations without the 'reference depth subtraction' in Eq. 3, so as to verify what is helping the system? the 'inner-depth' with reference depth subtraction, or enforcing depth in the *softmax averaged* fashion instead of D-bin classification?
> >
> > (p.s. some of the notations in the updated Sec 3.2 are redundant and overall it is very difficult to read. e.g., $S_j$ and $D_i$ seem to be from the same depth estimation but only with different subscripts. )
> >
> > Q2. the reviewer appreciates the newly added references to existing works. with that said, he still feels that this paper introduces very limited new knowledge in terms of the distillation method itself (Eq. 5-8 are all direct applications of [r2]).

---

> > > ### Author Response · Authors · 2023-11-22
> > > **Response to Feedback by Reviewer oss3**
> > >
> > > Thank you very much for your quick reply. We hope our further response can address your concerns.
> > >
> > > **Q1.** Sorry for the caused confusion. In the description "The introduction of relative depth is different from normalization which amplifies the differences between similar depth clusters in a certain area", this clause ("amplifies the differences between similar depth clusters in a certain area") is used to refer to normalization, not inner-depth supervision. We did not perform normalization on the foreground objects, nor did we achieve a similar effect to normalization.
> > >
> > > It is noteworthy that the inner-depth supervision is not just a trick, but poses insight on ***how to select the optimal reference point adaptively for different objects?*** This is because the naive subtraction of a large reference value not always help the network to focus on the nuances as expected. We have shown the ablation study of inner-depth supervision in Table 4. We compare different settings for relative depth value calculation and depth reference selection. If the subtraction of a large reference value can bring improvements, both 2D center or 3D center reference points would benefit a lot from it, but it doesn't seem to be the case based on the experimental results. Thus, it can be seen that this method is not equivalent to normalization and does not achieve the effect of normalization.
> > >
> > > We have conducted many experiments for exploration and finally selected the pixel with the smallest depth prediction error as the
> > > reference point for each target, correspondingly set its depth value as the depth reference. As shown in Table 4, our adaptive reference obtains the best improvement, which indicates the dynamic depth reference point can flexibly adapt to different targets for inner-geometry learning.
> > >
> > > |Depth Reference |mAP $\uparrow$|NDS $\uparrow$|
> > > |--- | --- | --- |
> > > |-   |35.9|45.4|
> > > |3D Center  |35.8|45.2|
> > > |2D Center  |35.8|45.2|
> > > |**Adaptive Reference**|**36.6**|**46.1**|
> > >
> > > #### Note: The results in the table all use $\mathcal{L}^A_{\rm{depth}}$ and $\mathcal{L}_{\rm{bev}}$ by default.
> > >
> > > The adaptive reference point is not like the 'mean' but a real reference for relative relationships inside the foreground objects. The key insight lies in that absolute depth supervision ignores relationships of inner-depth and this often leads to abnormal depth predictions, such as sudden and discontinuous changes in depth prediction values between adjacent pixels within the same object. With the help of inner-depth supervision, we can get a better depth distribution of target objects.
> > >
> > > We notice that there is an extra concern, the key point of this issue is whether the supervision of continuous depth values is superior to D-bin classification. Regarding this issue, we are sorry for the limited time to do more ablations, but this concern has been addressed in the ablation study (Table 5) of BEVDepth.
> > >
> > > |Depth Loss |mAP $\uparrow$|NDS $\uparrow$|
> > > |--- | --- | --- |
> > > |BCE (D-bin classification)  |32.2|36.7|
> > > |L1 (continuous depth value)  |32.1|37.1|
> > > |BCE + L1|32.3|37.2|
> > >
> > > As shown in the table, there is not much difference in performance between these two methods, even combining them. The gain brought by our method does not come from the supervision of continuous depth values.
> > >
> > > Regarding the notions, sorry for the caused confusion, we use **S** to denote the categorical depth prediction while using **d** to denote the continuous depth value for distinguishing. Thank you for your suggestion, we will reconsider the specific notions.
> > >
> > > **Q2.**
> > > The knowledge distillation method in our model is not a simple utilization of existing methods, but delicately incorporate 3D-specific characters for 3D object detection.
> > >
> > > - We mainly focus on the foreground area. From the perspective of BEV, there are a large number of background areas that are redundant and often affect the learning of teacher features. Therefore, we mainly focus on learning internal feature relationships from the object level, rather than the entire BEV feature map, which is different from traditional distillation methods.
> > >
> > > - We extract the BEV area of each object target and represent it by a series of keypoint features. This design can reduce computational burden and adapt to objects of different shapes. Moreover, we enlarge the box size for a little bit in the BEV space to cover the entire foreground area, e.g., object contours and edges. By these, such BEV keypoints can well represent the part-wise features and the inner-geometry semantics of foreground targets.

---

### Official Review · Reviewer_CeGw · 2023-11-01

**Soundness:** 3 good
**Presentation:** 3 good
**Contribution:** 2 fair
**Rating:** 6
**Confidence:** 3

**Summary:**

Recently, improving camera based 3D object detection by leveraging the LiDAR pure 3D information is get attention in the academia and industry. This paper belongs to this category. Overall, the paper proposed a new multi-view 3D detection via TARGET INNER-GEOMETRY LEARNING. Specifically, the authors proposed to use inner-depth supervision and inner-feature BEV distillation to boost the performance of mutli-view BEV detection. Benchmarked on the nuScenes dataset, it showed the better performance. Meanwhile, the authors conducted the ablation studies to make the work solid.

**Strengths:**

+ Good initiatives and motivation to resolve the issues in camera based 3D detection
+ Good presentation with clear illustration to show the proposed method and great description of the proposed method (although it has some typos)
+ Comprehensive results and ablation studies on the nuScenes datasets

**Weaknesses:**

I think the novelty and motivation of this paper is pretty clear and I really like the showcase in the Figure 1. However, I have several concerns in the experimental verification parts.

The first concern is the experimental dataset is pretty limited to nuScenes. I would like to see some experimental analysis on the Waymo One, KiTTI or 3D KiTTI dataset which is also the standard for the 3D detection.

To follow the first concern, some results in the paper is cherry picked, such as in Figure 7, the author wanted to show the improvement visually, however, in my opinion, it is pretty cherry picked. If the author could illustrate the performance of small and far object detection, it will be better.

Another big concern is the proposed method is only compared to the BEV4D/BEVDepth and its variances. The author did not list the latest method, such as Cross-Modality Knowledge Distillation Network for Monocular 3D Object Detection etc. Although the proposed method is different, these methods are in the same catrgory as the proposed method, that is L-> C. I would like to see the comparisons in this category.

**Questions:**

Please check the weakness part and address the questions there. Overall I hope the author could present clearly and solid in the experimental results to make the paper as a strong submission.

---

> ### Author Response · Authors · 2023-11-21
> **Response to Reviewer CeGw**
>
> We sincerely appreciate your recognition of our motivation, and your valuable feedback serves as a tremendous source of encouragement to us. We apologize for any concerns that may have arisen, and we are committed to addressing them.
>
> ### **Q1. Experimental analysis on other 3D dataset.**
>
> We are very sorry for the misunderstanding. We have conducted relevant experiments on the KITTI dataset before which can be found in Appendix B.2 (Table 10). With our proposed method, we can achieve a higher 3D AP compared to CMKD [1].
>
> ### **Q2. Illustrate the performance of small and far object detection.**
>
> Thank you for your suggestions. We can illustrate the performance of small and far object detection from two perspectives:
>
> - **For qualitative analysis**, there are more visualization results shown on Figure 9 in Appendix B.2. These supplementary visualizations also serve to demonstrate that our proposed method is capable of mitigating a range of issues in detection, including the reduction of false positives and ghosting objects, as well as the refinement of certain 3D locations and orientations of bounding boxes. These benefits also apply to small and far objects.
>
> - **For quantitative analysis**, we conduct some additional studies and take BEVDet4D-R101 for an example.
>   - We consider small objects such as *pedestrians, motorcycles, bicycles, traffic cones, and barriers*, and evaluate the performance of small object detection by comparing the mAP across these categories:
>
>     |Baseline|$\mathcal{L}^A_{\rm{depth}}$| $\mathcal{L}^R_{\rm{depth}}$ | $\mathcal{L}_{\rm{bev}}$ |mAP$_{pedestrian} \uparrow$|mAP$_{motorcycle} \uparrow$|mAP$_{bicycle} \uparrow$|mAP$_{traffic\ cone} \uparrow$|mAP$_{barrier} \uparrow$|
>     |--- | --- | --- | --- | --- | --- | --- | ---|---|
>     |BEVDet4D-R101|  | | |43.3|34.5|32.4|54.8|55.2
>     |BEVDet4D-R101| $\checkmark$| | |44.6|35.6|32.8|56.3|55.9
>     |BEVDet4D-R101| $\checkmark$| $\checkmark$||45.4|37.3|33.7|57.8|58.8|
>     |**BEVDet4D-R101**| **$\checkmark$**| **$\checkmark$**| **$\checkmark$**|**45.7**|**38.8**|**37.5**|**58.9**|**57.2**
>
>     As demonstrated in the table, our proposed method enables more accurate detection of small objects.
>
>   - We use the distance between the object and the ego in the ego coordinate system to filter out far objects, and evaluate the performance of far object detection by comparing the mAP across different distance ranges:
>
>     |Distance (m)|Baseline|$\mathcal{L}^A_{\rm{depth}}$| $\mathcal{L}^R_{\rm{depth}}$ | $\mathcal{L}_{\rm{bev}}$ |mAP $\uparrow$|NDS $\uparrow$|
>     |--- | --- | --- | --- | --- | --- | --- |
>     |[0,30)|BEVDet4D-R101|  | | |44.2|53.7|
>     |[0,30)|BEVDet4D-R101| $\checkmark$| | |46.6|55.6|
>     |[0,30)|BEVDet4D-R101| $\checkmark$| $\checkmark$||47.4|55.8|
>     |**[0,30)**|**BEVDet4D-R101**| **$\checkmark$**| **$\checkmark$**| **$\checkmark$**|**48.6**|**56.8**|
>     |[30,60)|BEVDet4D-R101|  | | |10.2|28.7|
>     |[30,60)|BEVDet4D-R101| $\checkmark$| | |10.3|28.5|
>     |[30,60)|BEVDet4D-R101| $\checkmark$| $\checkmark$||11.7|30.1|
>     |**[30,60)**|**BEVDet4D-R101**| **$\checkmark$**| **$\checkmark$**| **$\checkmark$**|**12.6**|**30.2**|
>
>     As shown in the table, our method can improve detection performance even for distant objects.
>
> ### **Q3. Compare with other LiDAR-to-Camera Methods.**
>
> Please refer to **General Response to All Reviewers**, we list some latest works in the table. We also compare with CMKD as said in **Q1**. We're sorry for any concern this may have caused you.
>
> ### Reference:
>
> [1] Hong Y, Dai H, Ding Y. Cross-modality knowledge distillation network for monocular 3d object detection[C]//European Conference on Computer Vision. Cham: Springer Nature Switzerland, 2022: 87-104.

---

> > ### Author Response · Authors · 2023-11-22
> > **Respectful Inquiry Before Discussion Deadline**
> >
> > Dear Reviewer CeGw,
> >
> > Thanks again for your great efforts and constructive advice in reviewing this paper! With the discussion period drawing to a close, we expect your feedback and thoughts on our reply. We put a significant effort into our response, with several new experiments and discussions. We sincerely hope you can consider our reply in your assessment. We look forward to hearing from you, and we can further address unclear explanations and remaining concerns if any.
> >
> > Regards,
> >
> > Authors

---

### Official Review · Reviewer_v25e · 2023-11-04

**Soundness:** 3 good
**Presentation:** 3 good
**Contribution:** 4 excellent
**Rating:** 8
**Confidence:** 4

**Summary:**

The paper presents a novel approach to 3D object detection, emphasizing the improvement of camera-based detectors. The proposed method introduces "Inner-depth Supervision" and "Inner-feature BEV Distillation" techniques to enhance the learning of spatial structures and depth perception. The main results show a comprehensive performance evaluation on the nuScenes test set, where the authors compare their method with established benchmarks across several metrics, demonstrating its effectiveness. The ablation study likely delves into the specific contributions of different method components, though the exact details weren't extracted here.

**Strengths:**

The manuscript presents innovative techniques such as "Inner-depth Supervision" and "Inner-feature BEV Distillation", indicating a high degree of originality by potentially filling a gap in the camera-based 3D object detection literature. The quality of research seems robust, as evidenced by comprehensive evaluations and methodical ablation studies. Clarity, while harder to fully assess without the complete text, appears to be aided by the use of illustrative figures and a structured presentation of methods and results. The significance of the work is underlined by its potential to enhance the practicality and cost-effectiveness of 3D object detection systems, which could have far-reaching implications for autonomous driving and robotics. This paper could represent a valuable contribution to the field, provided the results hold under peer review and the methods are as scalable and adaptable as implied.

**Weaknesses:**

To improve the paper, the authors could expand the methodology section, provide additional experimental results, and include more thorough comparisons with state-of-the-art techniques. Moreover, an in-depth discussion of the limitations and broader implications of the work would add value and show the authors' comprehensive understanding of their method's place in the field.

**Questions:**

Can you provide additional details on the mathematical formulation and implementation details of the "Inner-depth Supervision" and "Inner-feature BEV Distillation" techniques?
The paper mentions results on the nuScenes test set. Have you evaluated your method on other datasets or in varied real-world conditions to test its generalizability?

**Details Of Ethics Concerns:**

none.

---

> ### Author Response · Authors · 2023-11-21
> **Response to Reviewer v25e**
>
> Thank you for your positive and thorough review of our paper. We greatly appreciate you taking the time to provide such thoughtful feedback.
>
> ### **Q1: Comparisons with state-of-the-art techniques.**
>
> Thank you for your advice. We have made a comparison with other state-of-the-art techniques in **General Response to All Reviewers**. Besides, we also found that our method is not only highly competitive in 3D detection tasks, but camera-based models pre-trained with our method can also serve as better backbones for other 3D camera-only tasks, such as 3D occupancy prediction task.
>
> ### **Q2: An in-depth discussion.**
>
> Thank you for your valuable insight and we have added it in Appendix B.4.
>
> - One limitation that will inevitably be involved in depth-related explicit supervision is the ground truth from LiDAR. Due to the sparsity of LiDAR, both absolute depth supervision and relative depth supervision will be affected to some extent, though the performance of object detection will be better with inner-depth supervision under the same conditions. For sparse point clouds, a good way to alleviate it is through depth completion.
>
> - Another limitation is the occlusion problem of the target. Due to visual occlusion, the ground truth of the occluded object that we can obtain is limited, which will affect the learning of depth prediction and inner-depth learning. A good way to alleviate it is to consider multi-frame images for inner-depth supervision of the same object's interior, but a potential risk here is that if the intrinsic and extrinsic parameters are inaccurate, it will directly affect the coordinate system transformation between multiple frames, thereby affecting the ground truth of depth values.
>
> ### **Q3: Evaluated method on other datasets.**
>
> Thanks for your suggestion. We have evaluated our method on KITTI before and results can be seen in Appendix B.2 (Table 10).
>
> &nbsp;
> &nbsp;
> &nbsp;
> &nbsp;
>
> We also have further added details, refined our approach and conducted new experiments to address concerns raised by other reviewers. The revisions have improved the quality and clarity of our work. We hope that you will find our changes satisfactory and look forward to hearing your assessment of the revised manuscript.

---

> > ### Author Response · Authors · 2023-11-22
> > **Seek Further Discussions with Sincerity and Respect**
> >
> > Dear Reviewer v25e,
> >
> > Thanks again for your great efforts and constructive advice in reviewing this paper! With the discussion period drawing to a close, we expect your feedback and thoughts on our reply. We put a significant effort into our response, with several new experiments and discussions. We sincerely hope you can consider our reply in your assessment. We look forward to hearing from you, and we can further address unclear explanations and remaining concerns if any.
> >
> > Regards,
> >
> > Authors

---

### Author Response · Authors · 2023-11-21
**General Response to All Reviewers - Part 1**

Firstly, we would like to express our gratitude to all the reviewers for their hard work in reviewing and providing constructive feedback!

## Summary of modifications

In the latest version of our manuscript, we have made the following modifications:

- According to the suggestions of Reviewer CVGt, we refine the Fig 4,5,6 and correct some notions. Besides, we report resolution, backbone and update performance of recent works (e.g. UniDistill [1], X$^3$KD [2], DistillBEV [3]) in Tab 2.

- Add related works mentioned by Reviewer oss3 in Appendix A.

- Add Tab 8 in Appendix for a fair comparison with recent works.

- Add Tab 12,13 in Appendix to illustrate the performance of small and distant object detection.

- Add Limitations of Inner-depth Supervision in Appendix B.4 suggested by Reviewer v25e.

## Comparison with recent works

We noticed that most of the reviewers have the same concern about comparing with recent works, and we are so sorry for it.

In the latest manuscript, we have updated the comparison with recent and published works in Appendix B.2 (Table 8). We also show the table here for a comprehensive comparison with recent works (e.g. **UniDistill [1], X$^3$KD [2], DistillBEV [3]**). To ensure a fair comparison and clear understanding, we have also **included the baseline results reported by the authors in their papers**. When comparing the following methods, we use a **unified distillation mode of LiDAR-to-Camera**, evaluate them on NuScenes, and ensure that the resolutions are consistent across all methods.

 |Backbone |Student  | mAP $\uparrow$| NDS $\uparrow$|Method|mAP $\uparrow$| NDS $\uparrow$|Venue|
 |--- | --- | --- | --- | --- | --- | --- | --- |
 |ResNet50    | BEVDet     | 20.3   | 33.1   | UniDistill  | 26.0$_{(+5.7)}$  | 37.3$_{(+4.2)}$   | CVPR 2023
 |ResNet50    | BEVDet     | 30.5   | 37.8   | DistillBEV  | 32.7$_{(+2.2)}$   | 40.7$_{(+2.9)}$   | ICCV 2023
 |**ResNet50**|**BEVDet**  |**29.8**|**37.9**|**TiG-BEV**  |**33.1$_{(+3.3)}$**|**41.1$_{(+3.2)}$**|**Ours**
 |ResNet50    | BEVDet4D   | 32.8   | 45.9   | DistillBEV  | 36.3$_{(+3.5)}$   | 48.4$_{(+2.5)}$   | ICCV 2023
 |**ResNet50**|**BEVDet4D**|**32.2**|**45.1**|**TiG-BEV**  |**35.6$_{(+3.4)}$**|**47.7$_{(+2.6)}$**|**Ours**
 |ResNet50    | BEVDepth   | 35.9   | 47.2   | X$^3$KD$_{modal}$ | 36.8$_{(+0.9)}$   | 49.4$_{(+2.2)}$   | CVPR 2023
 |ResNet50    | BEVDepth   | 36.4   | 48.4   | DistillBEV  | 38.9$_{(+2.5)}$   | 49.8$_{(+1.4)}$   | ICCV 2023
 |**ResNet50**|**BEVDepth**|**35.7**|**48.1**|**TiG-BEV**  |**38.3$_{(+2.6)}$**|**49.8$_{(+1.7)}$**|**Ours**

 #### Note:

 #### (1) In this table, we show the performance evaluated in the validation set. Our proposed method is implemented by BEVDet_v1.0 [4] and the image resolution is 256 $\times$ 704.

 #### (2) The baseline performance of UniDistill is unexpectedly low, making it difficult to find an appropriate setting to compare with.

|Backbone |Student  | mAP $\uparrow$| NDS $\uparrow$|Method|mAP $\uparrow$| NDS $\uparrow$|Venue|
 |--- | --- | --- | --- | --- | --- | --- | --- |
 |Swin-B    | BEVDepth   | 48.9   | 59.0   | DistillBEV  | 52.5$_{(+3.6)}$   | 61.2$_{(+2.2)}$   | ICCV 2023
 |**ConvNeXt-B**|**BEVDepth**|**49.1**|**58.9**|**TiG-BEV**  |**53.2$_{(+4.1)}$**|**61.9$_{(+3.0)}$**|**Ours**

 #### Note: In this table, we show the performance evaluated in the test set. Our proposed method is implemented by BEVDet_v1.0 [4] and the image resolution is 640 $\times$ 1600. We don't use TTA and extra data.

As shown in tables, our proposed method achieves comparable performance among the latest state-of-the-art methods, **even with stronger backbones**.

---

> ### Author Response · Authors · 2023-11-21
> **General Response to All Reviewers - Part 2**
>
> ## Differences among recent works (1)
>
> Compared with methods of recent works, we would like to claim that:
>
> **(1) Our motivation mainly focuses on a better LiDAR-to-Camera learning scheme to effectively leverage LiDAR modality from both the low level (inner-depth supervision) and the high level (inner-feature supervision), which well relieves the modality gap.**
>
> **(2) Our inner-depth provides fine-grained depth cues for object-level geometry understanding, which can implicitly benefits BEV representation learning.**
>
> **(3) We pay attention to the relationships inside the foreground objects by inner-feature supervision, while most of the existing work focuses on BEV feature alignment and response distillation, similar to BEVDistill.**
>
> - For BEV feature distillation, most existing methods choose to use strict feature alignment. In our paper, we illustrate that forcing alignment of features from two different modalities is not ideal, as **it will be affected by modality differences**.
>
> - Some techniques like response distillation are not the main focus of our motivation, so we did not specifically use them although the improvement is promising.
>
> - We completely abandon strict feature imitation and instead learn the knowledge contained in the LiDAR-based teacher through modeling the internal relationships of foreground targets, allowing the camera-based student model to learn a better BEV representation.
>
> **(4) Our extensive experiments have shown that the improvement brought by feature alignment is limited, while distillation through learning inner-feature relationships is superior.**

---

> > ### Author Response · Authors · 2023-11-21
> > **General Response to All Reviewers - Part 3**
> >
> > ## Differences among recent works (2)
> >
> > The more detailed differences among these works as following:
> >
> > **(1) UniDistill** focuses on transferring knowledge from multi-modality detectors to single-modality detectors in a universal manner.
> >
> >   - Although UniDistill has shown us a good story for cross-modal distillation, its Feature Distillation and Response Distillation are very similar to those proposed by BEVDistill. What we need to declare is that response distillation is a promising method for improvement, but our motivation is not focused on this, so we did not specifically use it. As we mentioned in the table, there is a bit strange and makes it difficult to make a fair comparison due to the low baseline used.
> >
> >   - Though we do not recommend direct feature alignment, the feature alignment method proposed in this paper also has some confusing issues. Additionally, while the design of its Feature Distillation is intended to eliminate the influence of the background, the improvement is very limited if the background is not considered. Not to mention that in feature alignment, it only considers 9 key points instead of the overall bounding box area (which wouldn't even cost much computation).
> >
> >   - Our method considers the internal relationships of features from both spatial and semantic perspectives, rather than directly aligning features. Furthermore, we also consider the background area around the edges of the bounding box when sampling, which we found to be helpful for getting a clearer boundary. In addition, our inner-depth supervision can further help us extract the internal relationships of foreground objects, facilitate better inner-feature supervision and enhance the performance of camera-based detectors.
> >
> > **(2) X$^3$KD** is a knowledge distillation framework for multi-camera 3D object detection (3DOD), leveraging cross-modal and cross-task information by distilling knowledge from LiDAR-based 3DOD and instance segmentation teachers.
> >
> >   - In addition to commonly used feature distillation and response distillation in knowledge distillation (like BEVDistill), X$^3$KD also introduces adversarial training and explores the gains brought by cross-task distillation. It should be noted that these new modules (X-AT and X-IS) and our method are not in conflict. On the contrary, we can complement each other. But in the table, we only compare our method with X$^3$KD$_{modal}$ for a fair comparison from the LiDAR-to-Camera distillation perspective.
> >
> >   - The X-AT proposed by X$^3$KD is also aimed at mitigating the impact of modality differences by optimizing feature representation from a more global feature similarity perspective. This is an interesting method, but the improvement is a little limited. We focus on the more fine-grained relationships within foreground objects, optimizing from low-level inner-depth supervision to high-level inner-feature supervision, and have achieved superior results as demonstrated by our method. This observation emphasizes the promising potential of the TIG-BEV distillation method.
> >
> > **(3) DistillBEV** involves feature imitation and attention imitation losses across multiple scales, enhancing feature alignment between a LiDAR-based teacher and a multi-camera BEV-based student detector.
> >
> >   - In fact, DistillBEV is more like finely tuned feature alignment that utilizes multi-scale distillation methods. The multi-scale approach is a promising method to aid distillation. It is still directly aligning the features, and through multi-scale methods, making the feature alignment more strict. This paper to some extent reduces the harm caused by modal differences by selecting layers for distillation, but does not solve it better.
> >
> >   - The main difference between our method and others lies in motivation. Our method is straightforward and clear, achieving better results than feature alignment by learning the internal relationships of objects, which helps to alleviate differences between modalities while benefiting from the LiDAR modality.
> >
> >
> > ### Reference:
> >
> > [1] *Zhou S, Liu W, Hu C, et al. UniDistill: A Universal Cross-Modality Knowledge Distillation Framework for 3D Object Detection in Bird's-Eye View[C]//Proceedings of the IEEE/CVF Conference on Computer Vision and Pattern Recognition. 2023: 5116-5125.*
> >
> > [2] *Klingner M, Borse S, Kumar V R, et al. X3KD: Knowledge Distillation Across Modalities, Tasks and Stages for Multi-Camera 3D Object Detection[C]//Proceedings of the IEEE/CVF Conference on Computer Vision and Pattern Recognition. 2023: 13343-13353.*
> >
> > [3] *Wang Z, Li D, Luo C, et al. DistillBEV: Boosting Multi-Camera 3D Object Detection with Cross-Modal Knowledge Distillation[C]//Proceedings of the IEEE/CVF International Conference on Computer Vision. 2023: 8637-8646.*
> >
> > [4] *https://github.com/HuangJunJie2017/BEVDet/tree/master*